# Affinity-optimizing enhancer variants disrupt development

Fabian Lim[1,2,3,5], Joe J. Solvason[1,2,4,5], Genevieve E. Ryan[1,2,5], Sophia H. Le[1,2], Granton A. Jindal[1,2], Paige Steffen[1,2], Simran K. Jandu[1,2] & Emma K. Farley[1,2 ✉]

Enhancers control the location and timing of gene expression and contain the majority of variants associated with disease[1–3]. The ZRS is arguably the most well-studied vertebrate enhancer and mediates the expression of *Shh* in the developing limb[4]. Thirty-one human single-nucleotide variants (SNVs) within the ZRS are associated with polydactyly[4–6]. However, how this enhancer encodes tissue-specific activity, and the mechanisms by which SNVs alter the number of digits, are poorly understood. Here we show that the ETS sites within the ZRS are low affinity, and identify a functional ETS site, ETS-A, with extremely low affinity. Two human SNVs and a synthetic variant optimize the binding affinity of ETS-A subtly from 15% to around 25% relative to the strongest ETS binding sequence, and cause polydactyly with the same penetrance and severity. A greater increase in affinity results in phenotypes that are more penetrant and more severe. Affinity-optimizing SNVs in other ETS sites in the ZRS, as well as in ETS, interferon regulatory factor (IRF), HOX and activator protein 1 (AP-1) sites within a wide variety of enhancers, cause gain-of-function gene expression. The prevalence of binding sites with suboptimal affinity in enhancers creates a vulnerability in genomes whereby SNVs that optimize affinity, even slightly, can be pathogenic. Searching for affinity-optimizing SNVs in genomes could provide a mechanistic approach to identify causal variants that underlie enhanceropathies.

The human genome contains millions of enhancers[7]. These segments of the DNA act as switches to regulate where and when all the protein-coding genes in our genome are expressed. As such, enhancers encode the instructions for tissue-specific gene expression and thus successful development, adult homeostasis and cellular integrity[8]. Most SNVs associated with phenotypic variation and disease are located in enhancers[1–3]. Pinpointing which SNVs in an enhancer contribute to disease is a huge challenge because these causal variants are often embedded within a sea of inert variants[1–3,9]. Our inability to pinpoint causal enhancer variants is a bottleneck in relating genotype to phenotype. Here we use mechanistic and generalizable principles governing enhancers to predict causal enhancer variants. Such an approach could enable systematic and scalable methods that harness the full potential of genomic data to improve human health.

To investigate the relationship between enhancer sequence and phenotypes, we focus on the ZRS enhancer[4]. This enhancer regulates the expression of *Shh* (*SHH* in humans) in the posterior of the developing limb buds in a region known as the zone of polarizing activity (ZPA), and is crucial for limb and digit development in vertebrates such as chicks, mice and humans[10–12]. This approximately 800-bp enhancer is highly conserved in sequence between mice and humans, and in both species it is located nearly 1 Mb away from the *Shh* promoter[4]. Although the ZRS is one of the most well-studied enhancers, how it encodes gene expression and how SNVs cause phenotypes are unclear. More than 30

SNVs in the ZRS found across vertebrates are associated with polydactyly and other limb defects such as tibial hemimelia (shortening of the tibia) (Supplementary Table 1). Several human families and mice have the same SNVs within the ZRS and show similar phenotypes, providing evidence of the robustness of polydactyly phenotypes across species and genetic backgrounds (Supplementary Table 1). The high degree of conservation in digit patterning and ZRS sequence across mice and humans makes the mouse an excellent system in which to study the genetic basis of polydactyly. Reporter assays analysing the effect of SNVs on ZRS enhancer activity in mice suggest that polydactyly is associated with gain-of-function (GOF) ectopic enhancer activity in the anterior limb bud. Eight human SNVs associated with polydactyly have been tested within the endogenous mouse locus, and studies suggest that four of these cause *Shh* GOF expression in the anterior limb bud and extra digits[5,13] (Supplementary Table 2). However, the mechanisms by which these SNVs alter enhancer function are poorly understood[5,13,14].

## Redundant low-affinity ETS sites regulate the ZRS

The ZRS is regulated by a combination of transcription factors, including HAND2, HOX, ETV4, ETV5, ETS-1 and GABPa[13,15–18]. Five annotated sites known as ETS1–ETS5 are involved in the transcriptional activation of *Shh* from the ZRS and bind to the transcription factors ETS-1 and GABPa[15]. Both ETS-1 and GAPBa are activated downstream of fibroblast

[1]Department of Medicine, University of California San Diego, La Jolla, CA, USA. [2]Department of Molecular Biology, Biological Sciences, University of California San Diego, La Jolla, CA, USA. [3]Biological Sciences Graduate Program, University of California San Diego, La Jolla, CA, USA. [4]Bioinformatics and Systems Biology Graduate Program, University of California San Diego, La Jolla, CA, USA. [5]These authors contributed equally: Fabian Lim, Joe J. Solvason, Genevieve E. Ryan. ✉e-mail: efarley@ucsd.edu

growth factor (FGF) signalling from the apical ectodermal ridge[19]. Deletion of all five ETS sites results in the complete loss of enhancer activity in the ZPA when tested by reporter assays in mice[15]. Deletion of individual sites has no effect on expression; however, deleting combinations of these sites leads to a significant reduction of expression within the ZPA[15]. These results show that the five ETS sites (ETS1–ETS5) are redundantly necessary for the activation of *Shh* expression in the ZPA. An emerging regulatory principle that governs enhancers—including ones regulated by ETS—is the use of suboptimal-affinity binding sites (also known as low-affinity or submaximal binding sites) to encode enhancer tissue specificity[20–22]. This principle has been studied mainly in invertebrates[20–23].

To investigate whether the vertebrate ZRS also adheres to this regulatory principle, we measured the relative affinity of the five ETS sites (ETS1–ETS5) using protein binding microarray (PBM) data for the mouse transcription factor ETS-1 (refs. 24,25). PBM measures the binding affinity of all possible 8-mer sequences for the transcription factor of interest to provide a direct measurement of binding[25]. A relative affinity is then calculated by comparing the signal of all 8-mers to the signal of the highest-affinity site, which has a score of 1.00 or 100%. The DNA binding specificities of ETS-1 and other class I ETS transcription factors are conserved in mice and humans (Extended Data Fig. 1). Therefore, the binding affinities measured for ETS-1 convey the binding affinities of other class I ETS transcription factors that are expressed within the limb bud and which might also bind to this locus, such as GABPa[15,17,24]. Although PBM is an in vitro measurement, the relative affinity defined by PBM shows a strong correlation with in vivo chromatin immunoprecipitation (ChIP) peak intensity across several datasets (Extended Data Fig. 2).

The five previously annotated and functionally validated ETS sites (ETS1–ETS5)[15] in the ZRS have suboptimal binding affinities, ranging from 0.26 to 0.39 relative to consensus (Fig. 1a). We identify a total of 19 putative ETS sites in the human ZRS, and 18 in the mouse ZRS, all of which have suboptimal affinity. Fifteen of these sites are conserved in location and affinity between human and mouse (Extended Data Fig. 3a). One of these conserved sites is a newly identified ETS-A site that has an extremely low affinity of 0.15. We confirm that this ETS-A sequence binds the transcription factor ETS-1 using an electrophoretic mobility shift assay (EMSA) (Fig. 1a and Extended Data Fig. 4).

## Human polydactyly SNVs subtly increase affinity

The ETS-A site lies in a region of the ZRS that is completely conserved between mice and humans (Extended Data Fig. 3b). Two human variants associated with polydactyly—the French 2 (334 T>G) and Indian 2 (328 C>G) variants—are located in the ETS-A site (Fig. 1b). The French 2 variant is found in a family that has preaxial polydactyly; it is incompletely penetrant because three out of four family members with this variant have an extra thumb[6]. Only one individual with the Indian 2 variant has been identified and has preaxial polydactyly[5]. Notably, both human variants cause a similar subtle increase in the relative affinity of the ETS-A site from 0.15 in the reference to 0.24 in French 2 and 0.26 in Indian 2. EMSA confirms that both variants bind to ETS-1 more strongly than does the wild-type (WT) ETS-A (Extended Data Fig. 4). We hypothesized that this slight 1.6-fold increase in the relative affinity of the ETS-A site could be causing polydactyly. Although both variants have been studied using LacZ reporter assays in mouse, these assays suggest that only the French 2 variant alters enhancer activity[5]. Neither of these human variants have been studied within the endogenous ZRS locus, and the mechanism by which they alter gene expression is unknown. Therefore, we first sought to determine whether mice with the French 2 and Indian 2 variants exhibit ectopic expression of *Shh* and preaxial polydactyly.

French 2 and Indian 2 homozygous mice show ectopic expression of *Shh* in the anterior of the hindlimb at embryonic day (E)11.75 (Fig. 1h,l).

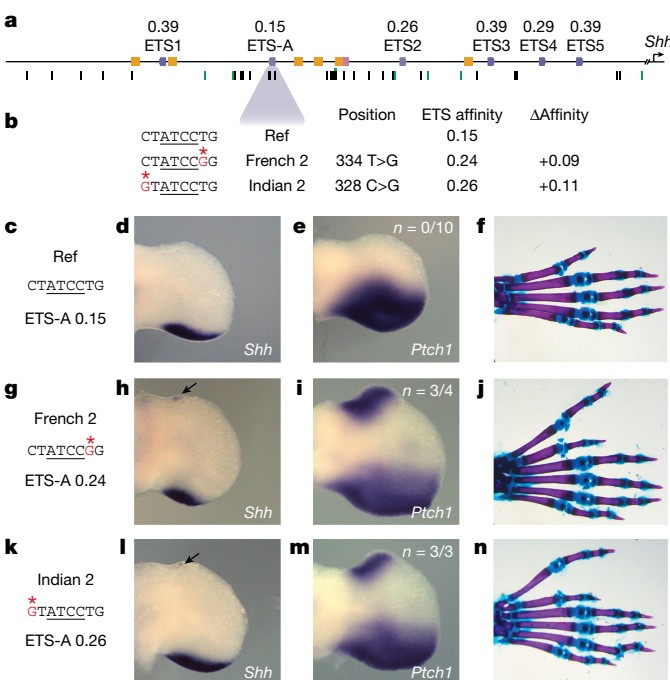

**Fig. 1 | An ETS-A site in the ZRS enhancer contains two human variants that are associated with polydactyly, both of which subtly increase ETS binding affinity. a**, The human ZRS contains five known and functionally validated ETS sites, ETS1–ETS5, all of which have suboptimal affinity. We identify a new site, ETS-A, which has a relative affinity of 0.15. Six HOX sites (yellow) and one HAND2 site (pink) have also been previously identified. Thirty-one SNVs associated with polydactyly are found in humans (black bars), and SNVs are also found in other species such as cats, mice and chicks (green bars). **b**, Two human SNVs associated with polydactyly, denoted French 2 and Indian 2, occur within the ETS-A site. Both SNVs lead to a subtle increase in relative affinity of ETS-A to 0.24 and 0.26 respectively. **c–f**, The ETS-A sequence in a reference (Ref) mouse (**c**) drives the expression of *Shh* (**d**) and *Ptch1* (**e**) restricted to the posterior domain of the developing limb bud in E11.75 and E12.0 embryos, respectively, as shown by in situ hybridization. **f**, Skeletal staining shows a WT mouse hindlimb with normal digit morphology. **g–j**, The French 2 SNV (**g**) drives the ectopic expression of *Shh* (**h**) and *Ptch1* (**i**) in the anterior limb bud of homozygous embryos (arrow) in addition to the normal domain of posterior expression. **j**, A mouse hindlimb homozygous for French 2 has an extra triphalangeal thumb. **k–n**, The Indian 2 SNV (**k**) drives the ectopic expression of *Shh* (**l**) and *Ptch1* (**m**) in homozygous embryos (arrow). **n**, A hindlimb from an Indian 2 homozygous mouse has an extra triphalangeal thumb. We did not calculate n for *Shh* because the expression is highly dynamic and thus hard to accurately capture; instead, we calculate the n of *Ptch1* as a readout of *Shh*.

The domain of ectopic *Shh* expression is tiny and highly dynamic. Therefore, we also looked at *Ptch1*, a direct downstream target of *Shh* that is commonly used as a readout for *Shh* signalling[13,26]. *Ptch1* is ectopically expressed in the French 2 and Indian 2 E12.0 homozygotes (Fig. 1i,m). We observed no ectopic expression of *Shh* or *Ptch1* in the forelimb at E11.75 or E12.0, owing probably to differences in the regulation of *Shh* expression in the forelimb and hindlimb[27]. Heterozygous and homozygous French 2 and Indian 2 mice have preaxial polydactyly in their hindlimbs, which indicates that very small and transient ectopic expression of *Shh* can have a strong effect on digit number (Fig. 1j,n). In humans, polydactyly occurs most commonly on the forelimbs, but in mice it typically occurs on the hindlimb[5,6,13] (Supplementary Table 1). This is likely to be due to differences in forelimb and hindlimb specification between the two species[28,29]. The additional anterior digit in mouse hindlimbs resembles the extra triphalangeal thumb observed in the orthologous human congenital malformations, and we call this a triphalangeal toe. Thus, both variants are causal for polydactyly and phenocopy the observed human phenotype.

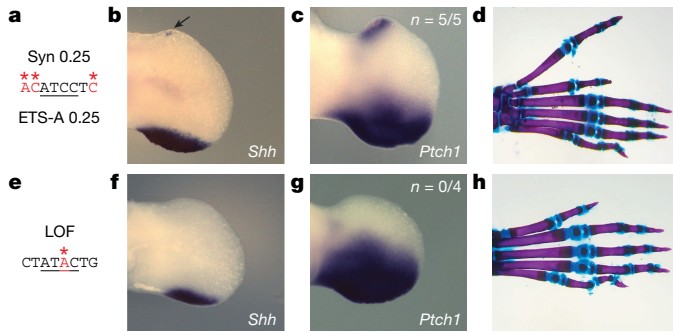

**Fig. 2 | Synthetic changes to the ETS-A site that create a 0.25-affinity site and a LOF site cause predicted phenotypes. a–d**, The Syn 0.25 ETS-A site (**a**), which has an affinity of 0.25, drives the ectopic expression of *Shh* (**b**) and *Ptch1* (**c**) in the anterior limb bud, in addition to the normal domain of expression in the posterior limb bud. **d**, Skeletal staining of a homozygous mouse hindlimb shows an extra triphalangeal thumb. **e–h**, The LOF ETS-A site (**e**) drives the normal expression of *Shh* (**f**) and *Ptch1* (**g**) in the posterior limb bud. **h**, Homozygous mice have normal digit morphology. We did not calculate *n* for *Shh* because the expression is highly dynamic and thus hard to accurately capture; instead, we calculate the *n* of *Ptch1* as a readout of *Shh*.

## Affinity-optimizing SNVs cause polydactyly

The French 2 and Indian 2 variants have the same phenotype, which suggests that the mechanism that drives polydactyly could be the same subtle increase in affinity of the ETS-A site. To test this prediction, we generated two more mouse lines with manipulations within the ETS-A site. The first mouse line, Syn 0.25, contains a synthetically created ETS-A site with an affinity of 0.25—the same affinity as that of the French 2 and Indian 2 variants—but has a different sequence change (Fig. 2a). We further validate the binding of ETS-1 to each of these ETS-A sequences with approximately 0.25 affinity using EMSA, and see no significant difference in the binding of ETS-1 to the French 2, Indian 2 or Syn 0.25 sequences (Extended Data Fig. 4). It is possible that any disruption to the ETS-A sequence could lead to a phenotype. To show that this is not the case, we also made a mouse line that we predicted would have no effect on phenotype. We created a loss-of-function (LOF) mouse in which the ETS-A binding site is ablated by removing a crucial nucleotide required for binding (Fig. 2e and Extended Data Fig. 4). Owing to the redundancy of ETS sites within the ZRS[15], we predicted that the loss of the ETS-A site would have no effect on *Shh* expression or limb development.

Mice containing the Syn 0.25 ETS-A site show ectopic expression of *Shh* and *Ptch1* in the anterior limb bud at E11.75 and E12.0, respectively, mirroring the expression patterns observed in the French 2 and Indian 2 mice (Fig. 2b,c). Syn 0.25 mice also have preaxial polydactyly in their hindlimbs (Fig. 2d), which suggests that the affinity change rather than the sequence change is driving the phenotype. As predicted, mice with the LOF mutation show no ectopic expression of *Shh* or *Ptch1* (Fig. 2f,g) and have normal limb morphology (Fig. 2h). Together, these studies show that the GOF increase in ETS-A affinity within the ZRS enhancer is pathogenic, whereas the LOF variant is non-pathogenic. This work demonstrates our ability to successfully predict the relationship between genotype and phenotype for sequence variants in the ETS-A site.

We hypothesized that if all three of these ETS-A affinity-optimizing variants share the same mechanism of action then the penetrance, laterality and severity of polydactyly should be comparable between the three lines. Phenotyping of mice was done blind to genotype. French 2, Indian 2 and Syn 0.25 mice with the same affinity ETS-A site have similar penetrance and laterality in heterozygotes, with polydactyly occurring most frequently on the right hindlimb (Fig. 3a). Homozygotes exhibit phenotypes bilaterally and have a higher penetrance of polydactyly

than do heterozygotes. Although all of the mice in this study are bred under identical conditions and have the same genetic background, there is a range of digit phenotypes (severity) in each line (Fig. 3b). Yet this distribution of digit phenotypes is identical across the lines. In heterozygotes the most common presentation is an extra digit that is either biphalangeal or triphalangeal, whereas homozygotes most commonly present with an extra digit that is triphalangeal. Thus, these three mouse lines with the same affinity increase show the same penetrance, laterality and severity with no statistical differences.

We next wanted to investigate whether the ETS-A site is indeed functional, which is challenging owing to redundancy within the enhancer[15]. If the WT ETS-A site is contributing to enhancer activity, then the penetrance of polydactyly in mice containing one copy of the ETS-A Syn 0.25 allele and one copy of the WT ETS-A site should be higher than it is in mice with one allele of ETS-A Syn 0.25 and one allele of LOF ETS-A. The penetrance and severity of phenotypes in the ETS-A Syn 0.25/LOF ETS-A mice are significantly reduced relative to the ETS-A Syn 0.25/WT ETS-A mice (Fig. 3), thus demonstrating that the 0.15-affinity WT ETS-A site contributes to enhancer activity and phenotypes.

Our comprehensive analysis of phenotypes across 795 transgenic mice shows that all three variants that increase the affinity of ETS-A to around 0.25 have indistinguishable phenotypes in heterozygotes and homozygotes. This provides compelling evidence that the subtle affinity optimization of this ETS-A site is the mechanism that drives polydactyly. Although there is increasing recognition of the role of low-affinity sites within enhancers, sites with affinities as low as 0.15, and even those with affinities of 0.25, are still typically ignored[5]. Yet here we see that a 0.15-affinity site is functional, and that a 0.25-affinity site is not only functional but sufficient to disrupt normal limb development, indicating that subtle increases in low-affinity sites can be pathogenic.

## Predicting penetrance and severity

Having seen that a subtle increase in affinity can cause developmental defects, we wondered whether the degree of affinity change could predict the penetrance and severity of phenotypes. This mechanistic understanding could be valuable for diagnostic and treatment purposes. To test this hypothesis, we created a mouse line with a 0.52-affinity ETS-A site (Fig. 4f). EMSA confirms that this sequence binds ETS-1 more strongly than do the WT ETS-A or the 0.25 ETS-A sites (Extended Data Fig. 4).

French 2, Indian 2 and Syn 0.25 homozygotes have a small amount of ectopic *Shh* expression in the hindlimb, and the Syn 0.52 site, as predicted, causes a large domain of *Shh* and *Ptch1* expression in the anterior of the hindlimb, as well as ectopic expression in the forelimb (Fig. 4g and Extended Data Fig. 5g,i). Consistent with these expression patterns, Syn 0.52 mice have polydactyly in both the forelimbs and the hindlimbs, whereas mice with 0.25 affinity have polydactyly only in the hindlimbs (Fig. 4h,i). Moreover, in Syn 0.52 mice, polydactyly is fully penetrant in the hindlimbs for both heterozygotes and homozygotes, and is almost fully penetrant in the forelimbs (Extended Data Fig. 6a). Polydactyly in Syn 0.52 mice is most commonly bilateral in both heterozygotes and homozygotes, whereas unilateral phenotypes are prevalent in 0.25 ETS-A affinity mice. Most Syn 0.52 mice have six digits that are all triphalangeal, but some have seven or even eight digits; this is more severe than the polydactyly seen in the 0.25 ETS-A affinity mice (Extended Data Fig. 6c). Syndactyly also occurs more frequently in the Syn 0.52 mice than in the French 2, Indian 2 and Syn 0.25 mice. In addition, Syn 0.52 mice have defects in the long bones. Tibial shortening or tibial hemimelia, a condition seen in humans[13], occurs in 95% of Syn 0.52 homozygotes (Fig. 4j and Extended Data Fig. 6b). Thus, as predicted, greater increases in affinity lead to more penetrant and severe phenotypes. This raises the possibility that affinity increases could be used to predict the severity and penetrance of phenotypes.

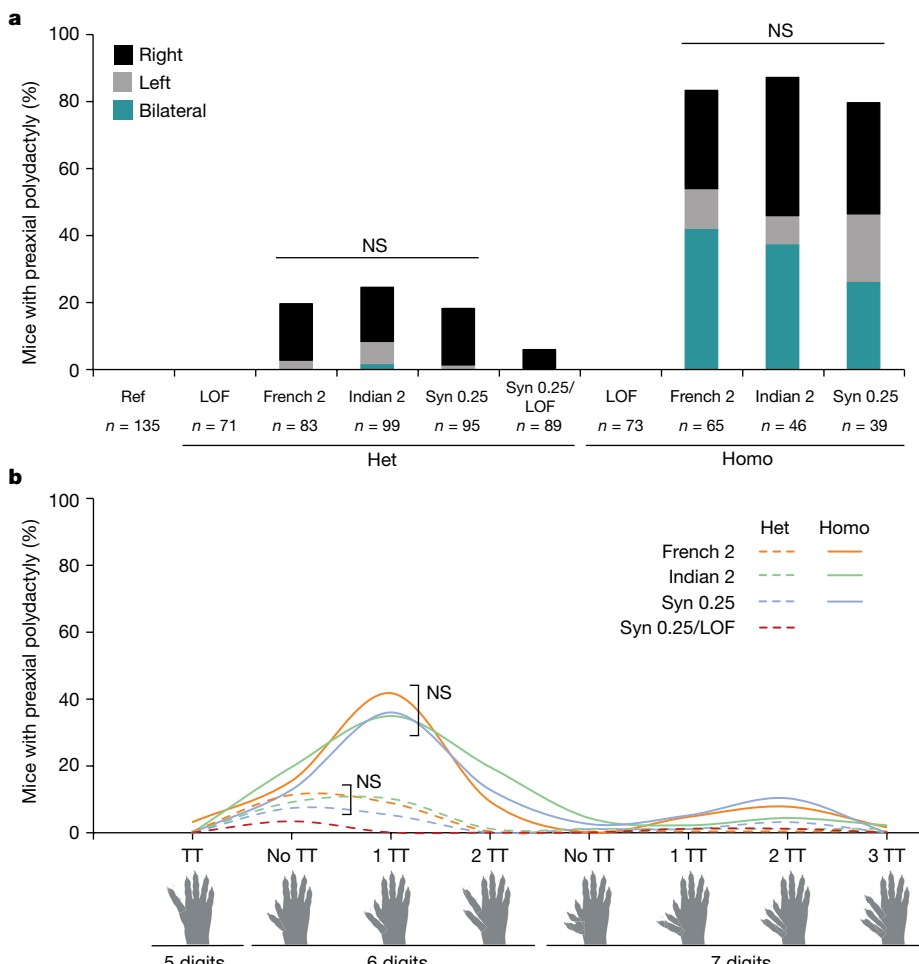

**Fig. 3 | All mice with the approximately 0.25-affinity ETS-A are indistinguishable in terms of the penetrance, laterality and severity of polydactyly. a**, Penetrance and laterality of phenotypes seen in French 2, Indian 2 and Syn 0.25 heterozygous (Het) and homozygous (Homo) mice. There is no significant difference in penetrance and laterality between heterozygotes and between homozygotes of French 2, Indian 2 and Syn 0.25 mice (Fisher's exact test). Polydactyly occurs more frequently on the right hindlimb in both heterozygotes and homozygotes. In Syn 0.25/LOF mice, polydactyly is less penetrant than in heterozygous Syn 0.25 mice (Syn 0.25/WT). NS, not significant. **b**, Polydactyly phenotypes seen in French 2, Indian 2 and Syn 0.25 heterozygotes (dashed lines) and homozygotes (solid lines), as well as in Syn 0.25/LOF mice. There is no significant difference in the phenotypes seen between the French 2, Indian 2 and Syn 0.25 lines (Fisher's exact test). TT, triphalangeal toe. Biorender. com was used to generate images in this figure.

## Affinity-optimizing SNVs prevalent across the ZRS

We next wanted to see whether our ability to predict causal enhancer variants could generalize to the other ETS sites within the ZRS. Seventeen SNVs cause an increase of at least 1.6-fold in ETS affinity; this is the fold change occurring in French 2, Indian 2 and Syn 0.25 (Fig. 5a). To see whether these variants drive gain of function, we analyse data from published ZRS reporter assays. In one of these assays that tests the effect of 2% sequence mutagenesis of the human ZRS enhancer on expression in mouse limb buds[5], we find that enhancer variants containing affinity-optimizing ETS SNVs are significantly enriched for GOF expression. However, there are around 16 bp changes within each enhancer, so we cannot definitively attribute the GOF expression to an individual SNV (Extended Data Fig. 7).

To more accurately relate SNVs to changes in gene expression, we analysed a massively parallel reporter assay (MPRA) that conducted saturation mutagenesis of a 485-bp region of the ZRS enhancer in a limb-like cell line[30]. Because spatial expression cannot be assayed in a cell line, levels of reporter expression serve as a readout of GOF enhancer activity. There is a significant enrichment of GOF gene expression in enhancers that contain affinity-optimizing ETS SNVs, relative to all other SNVs within the dataset (Fig. 5b). By contrast, there is no significant enrichment of GOF activity within SNVs that do not alter ETS binding affinity. Four of the eleven affinity-optimizing SNVs in this region of the ZRS drive GOF gene expression; these SNVs occur in ETS-A, ETS-B, ETS2 and ETS3. In this cell-line MPRA, the French 2 and Indian 2 SNVs did not significantly increase expression[30]. The false negative classification of French 2 and Indian 2 is likely to be due to the use of a cell line and the difficulty of assessing dynamic and very subtle changes in gene expression. The Indian 2 SNV was similarly misclassified in another reporter assay within the developing limb bud[5], in which three embryos were screened for expression at E11.5. Our analysis of data from these two ZRS mutagenesis studies[5,30], and our in-depth study of variants within the ETS-A site, show that affinity-optimizing SNVs in four different ETS sites across the ZRS can cause GOF gene expression.

To see whether our findings generalize beyond ETS to other transcription factor binding sites (TFBSs) in the ZRS, we searched for HOX affinity-optimizing SNVs associated with polydactyly. Both HOXA13 and HOXD13 are expressed in the distal limb bud, and changes in their expression can affect digit and limb development[31]. The Dutch 2 SNV (165 A>G[32]) increases the affinity for both HOXA13 and HOXD13. EMSA confirms that the Dutch 2 SNV binds these HOX proteins more strongly than the WT sequence does, suggesting that an increase in HOX binding

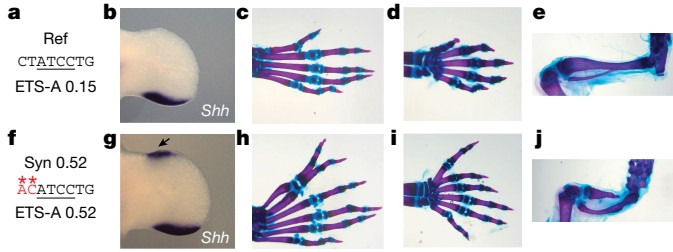

**Fig. 4 | A greater increase in affinity at the ETS-A site causes more severe and penetrant polydactyly and long-bone defects. a**, The ETS-A reference (Ref) sequence. **b**, In situ hybridization of *Shh* in WT hindlimb at E11.75 shows a domain of expression in the posterior of the limb bud. **c**, Skeletal staining of a WT mouse hindlimb showing five digits. **d**, WT forelimb showing five digits **e**, WT hindlimb focused on tibia and fibula. **f**, The Syn 0.52 ETS-A sequence. **g**, In situ hybridization of *Shh* in a Syn 0.52 homozygous mouse hindlimb bud at E11.75. Ectopic expression in the anterior of the limb bud is indicated by an arrow. **h**, Skeletal staining of a homozygous Syn 0.52 ETS-A mouse hindlimb showing six digits. The toe and the extra digit are both triphalangeal. **i**, Homozygous Syn 0.52 mouse forelimb showing six digits. Both the thumb and the extra digit are triphalangeal. **j**, Hindlimb focused on the tibia and fibula in a Syn 0.52 ETS-A homozygous mouse. The tibia is severely shortened. The hindlimbs in **e,j** were imaged with the same magnification. See Extended Data Fig. 6 for more details.

underlies the Dutch 2 polydactyly phenotype (Extended Data Fig. 8a,b and Supplementary Table 3).

## The enhanceosome contains affinity-optimizing SNVs

The interferon-β (IFNβ) enhanceosome is a well characterized enhancer that switches on IFNβ gene expression as an immune response to viral infection[33,34]. IRF-binding sites are necessary for enhancer activity[35]. We analysed MPRA assays that mutated every base pair within this enhancer[36]. SNVs that increase the affinity of IRF binding sites by a fold change of at least 1.5 are significantly enriched in GOF expression relative to all other SNVs in the MPRA, whereas there is no significant enrichment for GOF expression in SNVs within IRF sites that did not alter IRF binding affinity (Fig. 5c). One of these IRF affinity-optimizing SNVs was independently tested in another assay and also showed GOF expression[35]. These findings provide further evidence of the general role of affinity-optimizing SNVs in GOF expression within another enhancer regulated by different transcription factors and active in different cells. Unlike the ZRS enhancer, the enhanceosome is a redundant enhancer[37,38]. Therefore, within the context of reporter assays, the principle of affinity optimization applies to two classic enhancers and examples of a redundant and a non-redundant enhancer.

## Other transcription factors and disease enhancers

To investigate the role of affinity-optimizing variants in other enhancers, we looked at MPRA assays conducted on 11 enhancers associated with a variety of diseases[30] (Extended Data Fig. 9). Saturation mutagenesis was used to assay the effect of each base on enhancer activity. Each MPRA was performed in a cell line relevant for each particular enhancer[30]. Because FGF signalling is important in many cell types and aberrations to FGF signalling are implicated in a variety of disease contexts, we did not filter for enhancers responsive to ETS, but simply looked across all 11 enhancers. The median ETS affinity within these enhancers is 0.12. Enhancers with affinity-optimizing ETS SNVs (≥1.6-fold) are significantly enriched for GOF expression, whereas SNVs that occur in ETS sites but do not change their affinity are not significantly enriched in GOF enhancer activity (Fig. 6a). There are many SNVs that slightly increase binding affinity and are associated with GOF enhancer activity, indicating that small increases in binding

affinity can contribute to GOF gene expression across a wide range of enhancers.

We also looked at AP-1, a transcription factor that is involved in many cellular processes, including differentiation and cell proliferation[39]. Because AP-1 is a commonly used transcription factor, we again did not filter for enhancers known to be regulated by AP-1, but simply looked at all 11 enhancers. Similarly, we find that SNVs in the MPRA assay that increase the affinity of AP-1 (≥1.5-fold) lead to GOF gene expression, whereas SNVs in AP-1 sites that do not alter affinity are not significantly enriched in GOF expression (Fig. 6b).

To see whether our findings generalize to other datasets, we analysed an MPRA screen that tested lymphoblastoid regulatory elements and variants within these elements that were identified in an expression quantitative trait locus (eQTL) study[40]. This study looked at the genomes and lymphoblastoid transcriptomes of 446 individuals of Yoruba and European descent, and individuals from the 1000 Genomes Project[41]. The genomes and mRNAs from these lymphoblastoid cells were analysed to correlate genomic variation with changes in gene expression. Top-associated genomic variants, or variants in linkage disequilibrium with these eQTL variants, were chosen to be tested in the MPRA. In total, the effects of more than 3,000 SNVs were measured by MPRA in the same cell line as the eQTL study. As predicted, we see a significant enrichment in GOF enhancer activity within affinity-optimizing SNVs for both ETS and AP-1. SNVs occurring within ETS or AP-1 that do not alter affinity are not enriched for GOF enhancer activity (Fig. 6c,d). Together, these analyses on two orthogonal MPRA datasets show that, for two different transcription factors, over a massive variety of contexts, affinity-optimizing SNVs sites are a common mechanism driving GOF gene expression in reporter assays.

MPRAs allow us to study the effects of variants on expression in the context of a reporter assay, whereas eQTL infers the effects of variants on target gene expression in the context of the genome. Of the seven ETS affinity-optimizing GOF SNVs identified in the lymphoblastoid MPRA, five are significant eQTLs. All five of these eQTLs are associated with an increase in target gene expression (Extended Data Fig. 10a). Therefore, the increased reporter expression seen in the MPRA correlates with target gene expression in the endogenous locus. Indeed, when we analyse the eQTL signal of all 2,663 eQTL variants in the study[40], we find that ETS affinity-optimizing SNVs are significantly enriched in GOF expression for target genes (Extended Data Fig. 10b). By contrast, eQTL variants in ETS sites that do not alter affinity have no significant enrichment in GOF expression for target genes.

Genome-wide, eQTL ETS affinity-optimizing SNVs show significant enrichment in the increased expression of target genes (positive beta values), whereas SNVs that do not alter the affinity of ETS sites show no enrichment in the increased expression of target genes[41] (Extended Data Fig. 10c). Furthermore, with the eQTL dataset, we find that higher fold changes of ETS show a more significant enrichment of GOF target gene expression (Extended Data Fig. 10d). We also see this enrichment for AP-1 (Extended Data Fig. 10e). Although not all changes in expression detected in eQTL analysis are direct, the enrichment we see suggests that affinity-optimizing SNVs drive the GOF expression of target genes in the endogenous locus and that this GOF activity is not buffered within the endogenous context.

## Regulatory principles predict causal SNVs

Enhancers are littered with variation; therefore, predicting which SNVs are causal is a major challenge in relating genotype to phenotype. Within the ZRS MPRA dataset, 14.5% of all SNVs drive GOF expression, whereas 36% of affinity-optimizing SNVs drive GOF gene expression. Searching for affinity-optimizing ETS SNVs in the ZRS increases our ability to find causal GOF enhancer variants by 2.5 times when considering the MPRA data alone (Fig. 6e). The MPRA misses the two GOF variants that we identified in this study (French 2 and Indian 2). Thus,

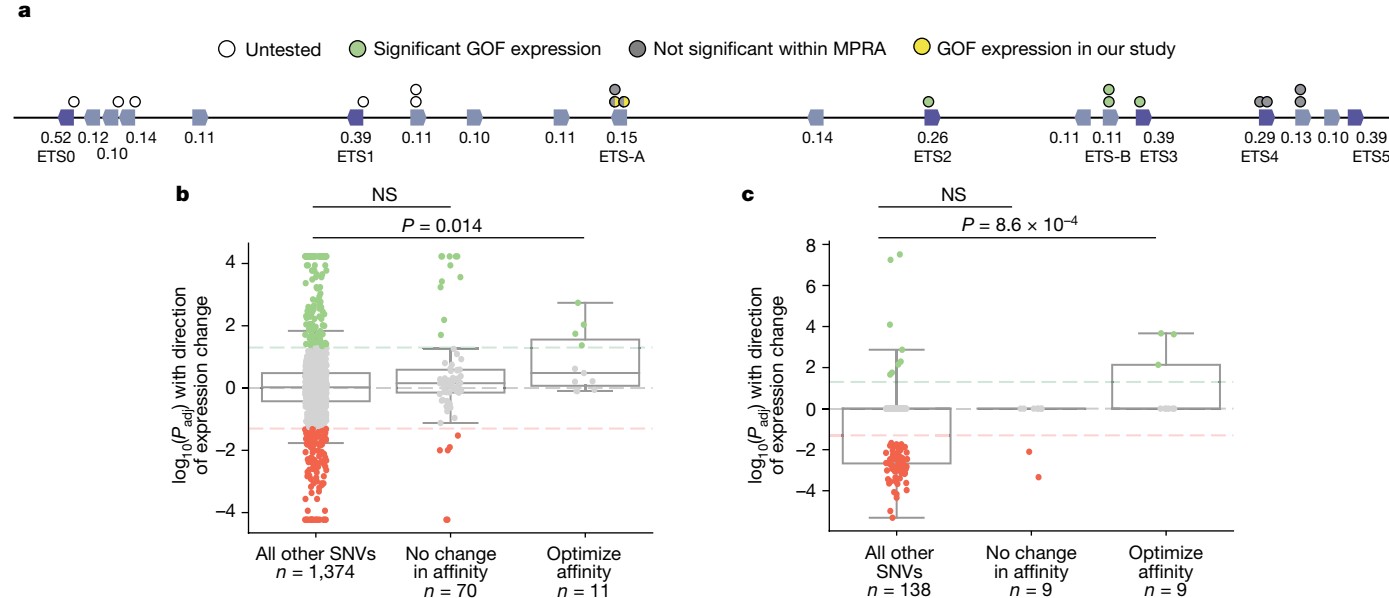

**Fig. 5 | Affinity-optimizing SNVs drive GOF expression in the ZRS and IFNβ enhanceosome. a**, Schematic of the human ZRS showing 19 putative ETS sites and 17 SNVs that increase the affinity of ETS sites by 1.6-fold or more. ETS sites identified from previous studies (ETS0–ETS5) are labelled[15,51]. We annotated changes in expression caused by these SNVs on the basis of results from a saturation mutagenesis MPRA and our study. **b,c**, Box plots showing all SNVs tested within MPRA mutagenesis experiments, their significance and their effects on expression. The bounds of the box plots define the 25th, 50th and 75th percentiles, and whiskers are 1.5× the interquartile range. The y axis is the log of the adjusted P value ($P_{adj}$) with the direction of expression change (positive values indicate an increase in expression and negative values indicate

a decrease in expression). Dashed horizontal lines indicate significance thresholds at $P = .05$. Each dot represents a tested SNV. SNVs that have a significant increase in expression are shown in green, those with no significant change in expression are grey and those with a significant decrease in expression are red. For each plot, we show SNVs that increase the affinity of the transcription factor, SNVs within the TFBS that do not alter affinity and all other SNVs. **b**, Analysis of the ETS TFBS in a saturation mutagenesis MPRA on a 485-bp region of the ZRS enhancer. **c**, Analysis of the IRF TFBS in the IFNβ enhanceosome MPRA on the IFNβ enhanceosome. We used one-tailed Mann–Whitney U tests to determine any significant enrichment for GOF enhancer activity.

when taking into account all known affinity-optimizing SNVs in the ZRS, 6/11 or 55% of affinity-optimizing ETS SNVs drive GOF expression, which increases our ability to find casual variants in the dataset by 3.8 times (Fig. 6e). Indeed, searching for SNVs that increase the affinity of ETS, AP-1 and IRF improves our ability to find causal GOF variants by twofold to sevenfold (Fig. 6e). Searching for affinity-optimizing SNVs is an effective and simple method to pinpoint putative causal enhancer variants, and could be a valuable filter to prioritize enhancer variants for further functional analysis.

## Discussion

Suboptimal-affinity binding sites are prevalent in a variety of enhancers, including Otx-a, svb, the ZRS, the IFNβ enhanceosome and many disease-associated enhancers and developmental heart enhancers[20–22,42]. Here we show that single-nucleotide changes that subtly increase binding affinity cause the loss of tissue specificity and organismal phenotypes in the mouse and human limb. In a complementary study, we find that affinity-optimizing SNVs in a *Ciona* heart enhancer drive ectopic expression of the gene *FoxF* in non-heart cells, which causes abnormal cell migration and heart defects as severe as two beating hearts[42]. Our results suggest that the prevalence of suboptimal-affinity sites creates a vulnerability in genomes whereby affinity-optimizing SNVs can drive ectopic GOF expression and phenotypes.

Given our findings, a greater focus on low-affinity but highly degenerate sites is essential to identify and predict casual enhancer variants. In this study, we show that PBM is a highly effective method to measure affinity. Although in vivo binding is no doubt modulated by other in vivo factors, such as protein–protein interactions, IDRs and other molecular interactions, our results show that the affinity of

transcription factor binding is a fundamental feature driving enhancer activity and gene expression. The use of such a simple in vitro measurement provides a systematic method to identify causal enhancer variants that does not rely on specific measurements within cell types. This enables a generalizable approach for pinpointing causal GOF enhancer variants that is applicable across genomes, cell types and even species.

The ETS-A site is an 8-bp sequence, and there are 65,536 possible sequence combinations within this region. Of these combinations, we successfully predicted the expression and phenotypes of five different ETS-A sequences (French 2, Indian 2, Syn 0.25, ETS-A LOF and Syn 0.52) on the basis of changes in binding affinity. These experiments demonstrate the power of mechanistic rules to predict causal enhancer variants. In the future, we will want to move beyond just identifying causal enhancer variants to predicting severity and penetrance. Within the ZRS ETS-A site, a greater increase in affinity causes more severe and penetrant phenotypes. This is likely to be true for all changes to affinity that occur in the same binding site and at the same position within an enhancer, because all of these affinity increases happen within the same context or grammar. The effects of affinity-optimizing SNVs are likely to be modulated by the surrounding binding sites, such that variants at different positions in an enhancer could have different effects despite having the same affinity increase[43]. We see evidence of this in our study of affinity-optimizing SNVs within the *Ciona* FoxF heart enhancer[42]. Integrating an understanding of affinity-optimizing SNVs and enhancer grammar will refine our ability to predict the severity and penetrance of enhancer variants.

We find that enhancer variants that cause GOF—but not LOF—enhancer activity disrupt development. The redundancy of enhancers on multiple levels ensures robustness within an organism. Typically, multiple enhancers known as redundant or shadow enhancers regulate

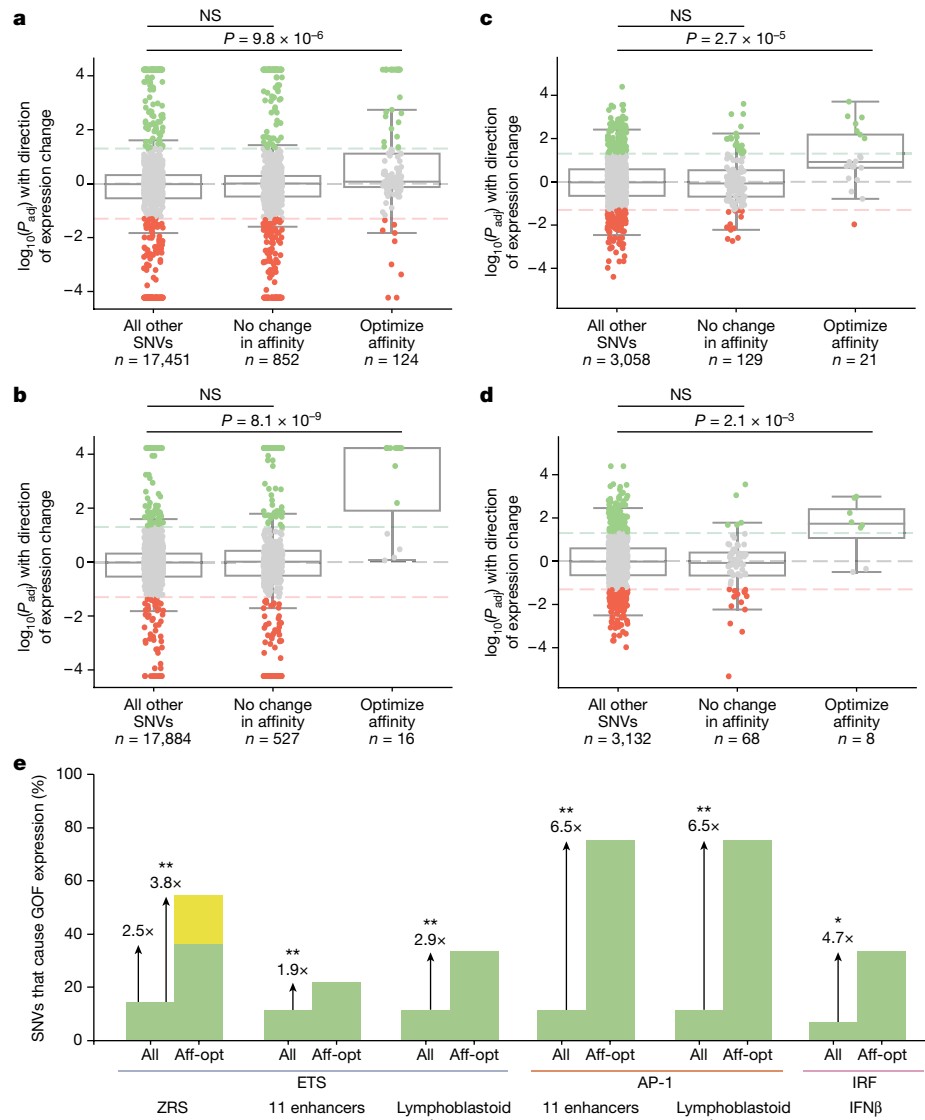

**Fig. 6 | Affinity-optimizing SNVs drive GOF expression in a wide variety of disease-associated enhancers. a–d**, Analysis of MPRAs for a variety of enhancers in different cell types. Box plots showing all SNVs tested within MPRA mutagenesis experiments and their significance and effects on expression. The bounds of the box plots define the 25th, 50th and 75th percentiles, and whiskers are 1.5× the interquartile range. One-tailed Mann–Whitney *U* test. **a,b**, Analysis of saturation mutagenesis MPRA assays of 11 disease-associated enhancers comparing all SNVs, SNVs within TFBSs that do not alter affinity and SNVs that increase the affinity of TFBSs for ETS (**a**) and AP-1 (**b**). **c,d**, Analysis of MPRA comparing the effect of SNVs within lymphoblastoid regulatory elements for the ETS TFBS (**c**) and the AP-1 TFBS (**d**). **e**, Filtering for affinity-optimizing

(aff-opt) SNVs significantly increases our ability to predict causal GOF enhancer variants. The bar graph shows the percentage of all SNVs that lead to GOF expression relative to the percentage of affinity-optimizing SNVs that lead to GOF expression. Green bars indicate SNVs that cause GOF expression within analysed MPRA datasets; yellow bar indicates SNVs that cause GOF expression in our the current study—namely, French 2 and Indian 2. Fisher's exact test was used to determine any significant enrichment for GOF expression in the all and aff-opt categories: **P < 0.001, *P < 0.01. Affinity-optimizing SNVs are those that lead to a fold change of at least 1.6 for ETS because this is the fold change for French 2, Indian 2 and Syn 0.25. Affinity-optimizing SNVs for AP-1 and IRF are those that cause a fold change of at least 1.5.

the same gene[44,45]. Another layer of enhancer redundancy is encoded within enhancers[46,47], as exemplified by the five ETS sites (ETS1–ETS5) in the ZRS[15]. Therefore, the loss of a single activator site, a reduction in the affinity of an activator site or even the loss of an entire enhancer can be compensated for by redundant sites or enhancers. By contrast, GOF variants that lead to increased levels of expression, or spatio-temporal ectopic expression, are harder to buffer and thus more likely to affect gene expression and development. The is exemplified by the LOF ETS-A site, which has no effect, whereas all four GOF variants drive ectopic gene expression and disrupt limb development. Focusing on variants that result in GOF expression could improve our ability to pinpoint causal enhancer variants.

Clusters of transcription factors are often found in close proximity to active enhancers. These have been described as hubs, and contain a large concentration of transcription factors that may be phase separated[48,49]. In such an environment, it is counterintuitive that single-base-pair changes can have such a marked effect on expression. At a biochemical level, we speculate that the subtle increase in affinity allows a longer dwell time for an activator; this could increase the chances of all required factors binding and the formation of a functional complex that can trigger transcription. Further investigations into the mechanisms by which one SNV that slightly increases binding affinity can nucleate transcriptional activation could help us to understand the driving forces behind transcriptional control.

Thirty-one human SNVs in the ZRS are associated with polydactyly. Only seven of these lie within validated binding sites; this is likely to be a result of the degenerate nature of binding sites and our poor annotation of functional binding sites in the ZRS. Of the seven SNVs in binding sites, three are affinity-optimizing SNVs. Two of these SNVs lie within the ETS-A site (French 2 and Indian 2) and one in a HOX site (Dutch 2). The ETS-A affinity-optimizing SNVs cause ectopic GOF expression in the anterior limb bud. Stem tetrapods were polydactylous with seven or eight digits[50]. The expression of repressors in the anterior limb bud might have contributed to the derived five-digit state. We speculate that the ETS-A affinity-optimizing SNVs cause ectopic expression in the anterior limb bud because the increase in affinity disturbs the balance of activators and repressors acting on the enhancer in the anterior limb bud, and because of an evolutionary sensitivity.

Millions of variants that are significantly associated with phenotypes and disease are located in enhancers[1–3], and functionally testing all of these is a major challenge. MPRA-style experiments assay the effect of enhancer variants through reporter assays; however, such assays have limitations as they do not test variants in the endogenous locus, or the appropriate cellular or multicellular context[30,40]. In addition, MPRAs tend to highlight variants that lead to the largest and most significant changes in expression. However, these might not have the greatest effect on phenotype. Indeed, the French 2 and Indian 2 variants drive ectopic expression in just a handful of cells for a short time, but this is sufficient for the formation of extra digits. Although not all small dynamic domains of ectopic expression will cause a phenotype, small changes in the temporal and spatial expression of morphogens, signalling-pathway proteins and effectors—especially in cell types in which these would alter identity—are likely to lead to patterning defects and phenotypes. Weighing the degree of expression change on the basis of the type of target gene and the sensitivity of the cellular context, and by focusing on variants that cause GOF rather than LOF expression, could improve our ability to predict enhancer variants that alter phenotypes.

Enhancers are often categorized according to several characteristics: their mode of interaction with the promoter; their level of sequence conservation; their distance from the target promoter; their target gene; the tissue in which they are active; and the species in which they are found. Our studies of the ZRS, FoxF heart enhancer, IFNβ enhanceosome and disease-associated enhancers, together with our eQTL analysis, show that the use of suboptimal-affinity sites to encode enhancer specificity, and the role of affinity-optimizing SNVs in GOF gene expression, transcend these categories. More broadly, the conservation of regulatory principles across diverse enhancers provides a framework for using violations of such rules to predict causal variants that underlie enhanceropathies.

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

## Methods

### Mice

All animal procedures and studies were approved by the University of California San Diego Institutional Animal Care and Use Committee according to the Association for Assessment and Accreditation of Laboratory Animal Care guidelines. Mice were maintained on a 12:12 light–dark cycle with ad libitum standard chow diet and water. Transgenic mouse assays were performed using *Mus musculus* C57BL/6 NHsd strain (Envigo). Animals of both sexes were included in this study.

### Generation of transgenic mice using CRISPR–Cas9

Cas9 protein (IDT, 1081058), trans-activating CRISPR RNA (tracrRNA) (IDT, 1072532), CRISPR RNA (crRNA) and single-stranded DNA (ssDNA) homology-directed repair template oligos were co-injected into one-cell embryos at the Moores UCSD Cancer Center Transgenic Mouse Shared Resource. Custom ssDNA repair oligonucleotides and crRNAs were synthesized by IDT. We designed and selected crRNA if the guide sequence is predicted to have high specificity on CRISPOR (http://crispor.tefor.net/crispor.py) and if the mutation introduced by homology-directed repair will ablate the protospacer adjacent motif (PAM) site. Because a PAM site is not available in the genomic locus for the human and synthetic mutations, we first generated a mouse line that contained a de novo PAM site within the ETS-A site. The French 2, Indian 2, Syn 0.25 and Syn 0.52 mouse lines were generated with CRISPR–Cas9, using one-cell embryos with the new PAM background (Supplementary Table 4). The LOF mutation mouse line was generated using embryos with the WT background. All mouse lines were generated by homology-directed repair using ssDNA as a repair template. Genome-edited founders were identified by genotyping as described below. Wherever possible, multiple founders bearing the same desired allele were used to establish each line. Founders were crossed to WT C57BL/6N mice to generate the $F_1$ generation for each mouse line.

### Genotyping

Genomic tail DNA was obtained and used to genotype ETS-A transgenic mice with the following primers: forward 5′-GGACAAGAGAT TAGCGTGGCTGGTGATTTCCTTTCACCCAGC-3′ and reverse 5′-GACACC AGACCAACTGGTAATGCATAATGACAGCAACATCC-3′. The underlined sequences anneal to the ZRS, and the remaining sequences are overhangs used to clone ZRS PCR products into a vector containing an ampicillin resistance cassette by Gibson assembly. For all mice, including founders, PCR products were analysed by Sanger sequencing (sequencing primer: 5′-CATCCTAGAGTGTCCAGAACC-3′) to identify ZRS genotypes. For all founder mice, PCR products were cloned, and individual clones were sequenced to confirm the initial genotyping results with single-allele resolution.

### Phenotyping

Each mouse born into our colony has all four limbs inspected by an investigator blind to genotype at postnatal day (P)10–18 during routine ear clipping (for identification) and tail biopsy collection (for genotyping). Limb and/or digit phenotypes, or the absence thereof, are readily detectable in postnatal mice and recorded in detail. Each limb is inspected for the number of digits and the presence of other features, including triphalangeal first digit(s) and/or shortened limbs. After genotyping, phenotypic data for each genotype in each ETS-A transgenic line are collated to calculate penetrance (on the basis of the presence or absence of phenotype).

### Statistical tests for mouse phenotypes

Fisher's exact test was used to measure any statistical difference in the penetrance and laterality of polydactyly across any pair of the approximately 0.25-affinity mouse lines (French 2, Indian 2 or Syn 0.25 mice). For penetrance, there are two factors: have phenotype or no phenotype. For laterality, there are three factors: bilateral, unilateral or no phenotype. To determine whether the occurrence of a unilateral phenotype has a bias on the left or right, $P$ values from chi square goodness of fit test were calculated. Fisher's exact test was used to measure any statistical difference in digit phenotypes across any pair of approximately 0.25-affinity mouse lines (French 2, Indian 2 or Syn 0.25 mice). $P$ values measuring any difference in digit phenotypes across males and females in each mouse line are also calculated. These tests have nine factors: five digits no TT, five digits one TT, … , seven digits three TT.

### Timed matings for embryo collection

Within each ETS-A transgenic mouse line, timed matings were set up and monitored each morning for vaginal plug formation. The date that plugs were observed was noted as E0.5. Females were removed from males on the plug date and embryos were staged at dissection. Embryos labelled as E11.75 have around 48 somites, and E12.0 embryos have around 53 somites. Pregnant females were humanely euthanized by isoflurane overdose. Embryos were dissected in ice-cold phosphate-buffered saline (PBS) pH 7.4 and then fixed in 4% paraformaldehyde in PBS pH 7.4 overnight with gentle rotation at 4 °C. Embryos were then dehydrated through a graded methanol series at 4 °C (25%, 50%, 75% methanol in PBS pH 7.4 plus 0.1% Tween-20, 100% methanol) and stored in 100% methanol at −20 °C for up to six months until use. The yolk sac of each embryo was collected and used for genotyping as described above. The sex of embryos is unknown.

### Probe cloning and synthesis for in situ hybridization

*Shh* and *Ptch1* templates were amplified from mouse E11.5 cDNA using previously described primers[52], and were ligated into a pCR BluntII TOPO vector, transformed into TOP10 competent cells and plated for selection on kanamycin plates. Colonies were selected for sequence verification and then plasmid prepped. Plasmid DNA was linearized with SpeI or NotI restriction enzyme and then used as a template for in vitro transcription using a digoxigenin labelling kit (Roche, 11175025910) with T7 (antisense) or Sp6 (sense) polymerase. After DNase treatment to digest template DNA, RNA probes were recovered using a RNeasy mini kit, and RNA concentration and purity were measured to confirm probe synthesis.

### Whole-mount in situ hybridization

Embryos were treated with 6% $H_2O_2$ in methanol for one hour, and then rehydrated through a methanol series to PBS-T (1% Tween-20 in PBS pH 7.4). Embryos were washed 5 × 5 min in PBS-T, and then treated with proteinase K (10 µg ml$^{-1}$) (Invitrogen, 1000005393) for 20 min. After permeabilization, embryos were washed in PBS-T containing 2 mg ml$^{-1}$ glycine, then PBS-T, then post-fixed for 20 min in 4% paraformaldehyde (PFA)/0.2% glutaraldehyde in PBS-T. Embryos were then washed 2 × 5 min in PBS-T, followed by 10 min in a 1:1 mixture of PBS-T and hybridization solution (50% formamide, 5× SSC pH 4.5, 1% SDS, 50 µg ml$^{-1}$ yeast tRNA, 50 µg ml$^{-1}$ heparin). Embryos were then allowed to sink (no rocking) in hybridization solution for 10 min. They were then changed to new hybridization solution and incubated for at least one hour at 65 °C. Hybridization solution was replaced with fresh hybridization solution containing 1 µg ml$^{-1}$ of antisense (all ETS-A embryos and WT control) or sense (WT control only) probe followed by overnight incubation at 65 °C. Embryos were washed 3 × 30 min in solution I (50% formamide, 5× SSC pH 4.5, 1% SDS) at 65 °C followed by 3 × 30-min washes in solution III (50% formamide, 2× SSC pH 4.5) at 65 °C. Embryos were then washed 3 × 5 min in TBS-T (1% Tween-20 in Tris-buffered saline) and blocked for 1 h in block solution (10% heat-inactivated sheep serum and 0.1% Roche blocking reagent in TBS-T). Roche blocking reagent (Roche, 11096176001) was dissolved in maleic acid buffer according to the manufacturer's recommendations.

Embryos were then incubated in block solution containing 1:2,500 anti-digoxigenin-AP antibody (Roche, 11093274910) overnight at 4 °C. Embryos were washed 3 × 5 min in TBS-T and then 5 × 1 h in TBS-T, followed by overnight incubation in TBS-T at 4 °C. Embryos were then washed 3 × 10 min in NTMT (100 mM NaCl, 100 mM Tris pH 9.5, 50 mM $MgCl_2$, 1% Tween-20) before coloration in AP reaction mix (125 µg ml$^{-1}$ BCIP (Roche, 11383221001) and 250 µg ml$^{-1}$ NBT (Roche, 11383213001) in NTMT). Coloration was carried to completion in the dark. Embryos were washed 10 min in NTMT followed by 3 × 10 min in TBS-T and then overnight in TBS-T at 4 °C. Embryos were imaged using the Leica M165 FC microscope with the Lumenera Infinity3 camera, then post-fixed in 4% PFA for 30 min and stored in 1% PFA in 4 °C. All steps were performed with gentle rocking and at room temperature unless otherwise specified.

## Skeletal preparations

Young postnatal mice at age P10-12 were humanely euthanized by $CO_2$ inhalation before skeletal preparations. Dissected limbs and/or whole cadavers of representative homozygotes for each line were skinned and eviscerated, then fixed in 95% ethanol overnight. Samples were then stained over two nights in cartilage staining solution (75% ethanol, 20% acetic acid and 0.05% Alcian blue 8GX (Sigma-Aldrich, A3157)), rinsed overnight in 95% ethanol, cleared overnight in 0.8% KOH and stained overnight in bone staining solution (0.005% Alizarin red S (Sigma-Aldrich, A5533) in 1% KOH). After staining, samples were further cleared in 20% glycerol in 1% KOH until digits were free of soft tissue and long-bone morphology was visible. Samples were further processed through a graded series of 50% and 80% glycerol in 1% KOH and then into 100% glycerol for imaging and storage. All steps of the skeletal staining procedure were performed with gentle rocking at room temperature.

## EMSA

EMSAs were performed using the LightShift Chemiluminescent EMSA Kit (Thermo Fisher Scientific) with biotinylated and non-biotinylated double-stranded oligonucleotides. Oligonucleotides were annealed according to an advanced protocol (https://tools.thermofisher.com/content/sfs/brochures/TR0045-Anneal-oligos.pdf). DNA-binding domain (DBD) proteins were synthesized using the TNT Quick Coupled Transcription/Translation System (Promega) from the pTNT plasmid for each respective protein. ETS-1 DBD (residues 336–441, which is conserved in sequence from human and mouse) was amplified from the pET28b-Ets1-ETS vector (Addgene, 85735). DBDs for human HOXA13 (residues 322–389) and HOXD13 (residues 276–335) were amplified from human genomic DNA, and sequences were confirmed by Sanger sequencing. The coding sequences for these DBDs were amplified with flanking XhoI and NotI sites and cloned into the pTNT-B18R vector (Addgene, 58978). The binding reaction was performed in a 20-µl volume containing 2 µl of 10× binding buffer (100 mM Tris, 500 mM KCl and 10 mM DTT; pH 7.5), 50 ng Poly(dI:dC), 20 femtomol biotin-labelled probe and protein extract. For competition experiments, a 200-fold molar excess of unlabelled probe was added. Binding reactions were pre-incubated for 10 min before adding the biotin-labelled probe. Binding reactions were then incubated at room temperature for 20 min and loaded onto a DNA retardation gel (6%). Electrophoresis with 0.5× TBE on ice and transfer to a 0.45-µm Biodyne B Nylon membrane (Thermo Fisher Scientific) was done in the cold room. DNA was cross-linked for 15 min using 312-nm light, and the membrane was put between blank sheets of paper overnight. The next day, the biotinylated probes were detected using the Chemiluminescent Nucleic Acid Detection Module (Thermo Fisher Scientific). Images of the resulting membrane were acquired using a Chemidoc MP imaging system (Bio-Rad). For quantification of ETS-1 binding to ETS-A variants, band quantifications were performed by taking the ratio of the volume (intensity) for the shifted band in the reaction with the ETS transcription factor to the volume (intensity) for the unshifted band in the reaction without the ETS transcription factor, using the Analysis Table in Image Lab 6.0.

## Calculation of binding affinity

Relative binding affinity is calculated using high-throughput binding data from the UniProbe database (http://thebrain.bwh.harvard.edu/uniprobe/index.php)[25,53]. Median intensity signals of 8-mers PBM data were measured as a percentage of their optimal 8-mer binding motif.

## Analysis of previously published MPRA data

MPRA data were downloaded from previously published papers[5,30,36,40]. Only single-base substitutions were considered across all datasets. We classified variants as significantly altering expression using $P$ values that were provided in the published supplementary tables. If the study did not report adjusted $P$ values, we adjusted all raw $P$ values using Benjamini–Hochberg adjustment. We defined variants as having a significant change in expression if their adjusted $P$ value was smaller than 0.05. We plot $\log_{10}(P_{adj})$ with direction of change in gene expression, in which positive values depict a significant variant that leads to increased gene expression. We used the one-tailed Mann–Whitney $U$ test to test for enrichments in GOF enhancer activity in different groupings (that is, 'All other SNVs', 'SNVs that do not change affinity' and 'SNVs that optimize affinity'). If there were more than 1,000 points in the dataset, we plotted a random 1,000 as dots over the box plots. MPRA data were analysed using standard Python libraries (pandas, numpy, scipy, seaborn, matplotlib). Processing, visualization and statistics were done using custom Python code.

ETS-binding sites were defined as NNGGAWNN[54]. We defined an ETS-optimizing variant as one that caused at least a 1.59-fold change (alt/ref) in all analyses with the exception of Extended Data Fig. 7. Analysis of the 2% sequence mutagenesis of the ZRS enhancer in Extended Data Fig. 7 defines ETS-optimizing variants as a fold change greater than 1.0, because only three variants within this dataset have a fold change of at least 1.59.

The AP-1-binding site was defined as NTKANNMA. IRF binding affinity was defined as NWNNGANA. Motifs used for ETS, AP-1 and IRF were determined on the basis of crystal structure and mutational analysis data[54–57]. We defined AP-1 and IRF-optimizing variants as one that caused at least a 1.5-fold change in binding affinity. We defined SNVs not changing the TFBS affinity as SNVs with a 0.8–1.25-fold change in affinity. For analyses on the IFNβ enhanceosome, we excluded nucleotides that contributed to two overlapping binding sites, and only analysed the effects of affinity-optimization SNVs that affect a single binding site.

## Analysis of eQTL data

We analysed eQTL data downloaded from the EBI eQTL catalogue GitHub page (https://github.com/eQTL-Catalogue/eQTL-Catalogue-resources/blob/master/tabix/) for a lymphoblastoid cell line generated by the Geuvadis consortium[41]. These eQTL data were used to design the MPRA library used previously[40,41]. For the eQTL data, we adjusted the raw $P$ values using the Bonferroni procedure, in which the total number of tests is the total number of genotype–gene-expression associations tested. For the seven ETS affinity-optimizing SNVs that cause significant differential expression in the MPRA experiments[40] and have eQTL values, we compare the MPRA expression and eQTL using adjusted log $P$ values plotted with the direction of differential expression for reporter assay and target gene expression (β value). For AP-1 affinity-optimizing SNVs that gave significant differential expression in the MPRA, only three overlapped with significant eQTL variants and so we did not study these. For larger-scale analysis of the relationship between significant variants and target gene expression, we analysed all eQTL variants in the MPRA library ($n$ = 2,663) in a previous report[40], and their effects on target gene expression (β values). Finally, for genome-wide analysis of all eQTL variants in lymphoblastoid

cell lines, we analysed all eQTLs from the Geuvadis consortium[41] with $P_{adj} < 0.01$. We categorized the eQTL variants into three categories: affinity-optimizing SNVs, SNVs that do not alter affinity and all other SNVs. For eQTL variants that have multiple genes associated, we plotted only the most significant association. A one-tailed Mann–Whitney $U$ test was used to determine any significant enrichment in eQTL variants that do not change affinity or increase affinity with GOF target gene expression (β value).

## PBM–ChIP correlation analysis

BigWig files for ChIP–seq data were downloaded from Gene Expression Omnibus accessions for previously published data[58–60]. BigWig files were chosen because they contain the most quantitative metric at base-pair resolution for the ChIP–seq signal. For each dataset, we predicted ETS sites using NNGGAWNN across the reference genome used to create the bigWig. We then extracted the average bigWig ChIP–seq signal over all predicted ETS TFBS 8-mers. We associated each ETS TFBS with its predicted ETS affinity using PBM[24]. We placed the ETS TFBS into bins of PBM affinity 0–0.1, 0.1–0.2, 0.2–0.3, … 0.9–1.0. Within each bin, we took the average bigWig ChIP–seq signal across each chromosome and plotted the chromosomal averages. The Spearman correlation uses all points from all bins and all chromosomes.

## Statistics

To assess any statistical differences in the penetrance and laterality percentages between French 2, Indian 2 and Syn 0.25 mice, we performed Fisher's exact test using the fisher.test function in R (Supplementary Table 3). Statistical differences in digit phenotypes were measured using Fisher's exact test using a 2 × 9 table (Supplementary Tables 3 and 4). To determine whether unilateral polydactyly deviates from the assumption that phenotype would occur at a 50%–50% rate on the right and left hindlimbs, a chi square goodness of fit test was used (Supplementary Tables 3 and 4). To measure any statistical difference between the band intensities in EMSAs across the French 2, Indian 2 and Syn 0.25 sequences, we performed a one-way ANOVA test and found no significant difference with $P = 0.18$.

## Reporting summary

Further information on research design is available in the Nature Portfolio Reporting Summary linked to this article.

## Data availability

All data supporting the findings of this study are available within the paper and its Supplementary Information. ChIP–seq data analysed in this paper were downloaded using accession codes GSM2218592, GSM3520734 and GSM4110116. eQTL data analysed in this paper were downloaded from the EBI eQTL catalogue GitHub page (https://github.com/eQTL-Catalogue/eQTL-Catalogue-resources/blob/master/tabix/) for the lymphoblastoid cell line generated by the Geuvadis consortium. The UniProbe database (http://thebrain.bwh.harvard.edu/pbms/UniPROBE_staging/browse.php) was used to access PBM data. Source data are provided with this paper.

## Code availability

All custom code used in the analyses has been deposited at GitHub (https://github.com/jsolvason/nature2023-limb) and Zenodo (https://doi.org/10.5281/zenodo.10368918).

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

**Acknowledgements** We thank all members of the E.K.F. laboratory for discussions; K. Cooper for help with identifying the ectopic *Shh* expression; the Moores UCSD Cancer Center Transgenic Mouse Shared Resource and in particular, J. Zhao for creating our transgenic mice; R. Tewhey for providing MPRA and eQTL data; K. Alasoo for providing eQTL data; M. D'Antonio for providing advice on eQTL analyses; A. D'Antonio-Chronowska, S. Mahesula, and S. Reck-Peterson for advice on EMSA experiments; and D. Higgs, J. Posakony, C. Tabin and M. Levine for reading our manuscript and providing feedback. The authors declare no competing interests. This study is supported by NIH DP2HG010013, NSF CAREER 2239957 and UC San Diego Start-up Funds. G.E.R. was supported by a UCSD Cancer Fellowship and NIH T32HL007444; J.J.S. was supported by NIH T32GM127235 and NIH DP2HG010013; E.K.F., S.H.L. and F.L. were supported by NIH DP2HG010013 and NSF CAREER 2239957; and G.A.J. was supported by NIH DP2HG010013 and NIH T32HL007444. Mice were generated at the Moores UCSD Cancer Center Transgenic Mouse Shared Resource supported by NIH P30 CA023100 and NIH P30 DK063491.

**Author contributions** G.E.R. generated mice, with support from E.K.F. F.L., G.E.R., S.H.L. and P.S. performed genotyping and phenotyping of mice. F.L. and G.E.R. performed in situ hybridization experiments and skeletal preparations. F.L. and S.H.L. conducted quantitative analysis of phenotypes. F.L. performed statistical tests on phenotypes. F.L. and G.A.J. performed EMSA. J.J.S. and S.K.J. performed PBM–ChIP correlation analysis. J.J.S. performed MPRA and eQTL analyses. E.K.F. devised the study. E.K.F., F.L., J.J.S. and G.E.R. designed the experiments; E.K.F., F.L. and J.J.S. wrote the manuscript. All authors provided feedback on the manuscript.

**Competing interests** The authors declare no competing interests.

**Additional information**
**Correspondence and requests for materials** should be addressed to Emma K. Farley.

|  | Helix 1 | Helix 2 | Helix 3 | Strand 4 |
|---|---|---|---|---|
|  | 1 2 | 3    4 | 5 6 7   8 9   1011  12 | 1314  15 |
| Human  ETS-1 | LWQF | WGKRK | YEKLSRGLRYYY | RYVY |
| Mouse  ETS-1 | LWQF | WGKRK | YEKLSRGLRYYY | RYVY |

**Extended Data Fig. 1 | Class I ETS family members have conserved DBDs.** DBDs of ETS-1 transcription factors. Numbers show the location of contacts between the DNA-binding site and the protein in human and mouse[24,61,62]. The DBDs are highly conserved across human, mouse and flies across other class I ETS family members[24,63,64].

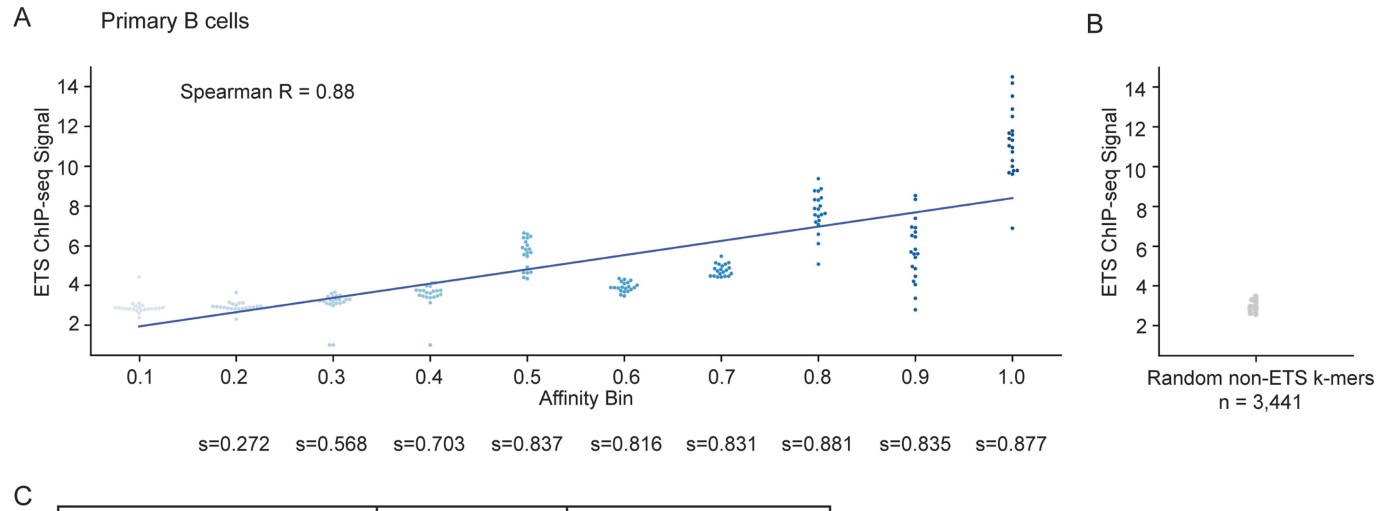

A  Primary B cells

Spearman R = 0.88

s=0.272   s=0.568   s=0.703   s=0.837   s=0.816   s=0.831   s=0.881   s=0.835   s=0.877

B

C

| Cell Type | Spearman R | Spearman R p-value |
|---|---|---|
| Primary B cells | sr = 0.88 | sp < 1e-5 |
| Natural killer cells | sr = 0.67 | sp < 1e-5 |
| THP-6 cells | sr = 0.81 | sp < 1e-5 |

**Extended Data Fig. 2 | PBM binding affinities correlate with the in vivo ETS-1 ChIP signal in various cell types. a**, ETS-1 ChIP–seq signals from mouse primary B cells (*y* axis) and PBM mouse ETS-1 affinities (*x* axis) show a strong Spearman's rank correlation of 0.88 (ref. 58). To show that the correlation is not heavily dependent on values from affinity bins 0.1 and 1.0, we show the Spearman's rank correlation between cumulative affinity bins below the graph. For example, the Spearman's correlation using bins 0.1, 0.2, 0.3, 0.4 is 0.703. **b**, The ETS-1 ChIP signal is low for all random non-ETS *k*-mers **c**, Correlation of ChIP signal and PBM affinity for other ETS-1 ChIP assays in mouse primary B cells, human natural killer cells and human THP-6 cell lines[58–60].

A

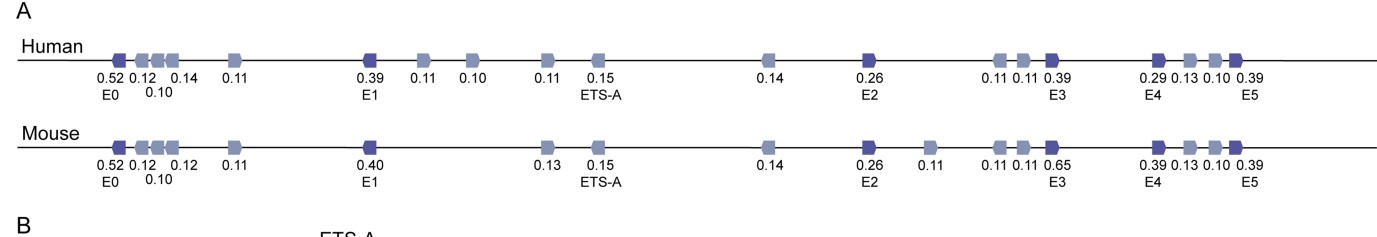

B

ETS-A

Human    TTTTAATATGTTTCTATCCTGTGTCACAGTTTGA
Mouse    TTTTAATATGTTTCTATCCTGTGTCACAGTTTGA
         ******************************

**Extended Data Fig. 3 | Conservation of ZRS ETS sites between humans and mice. a**, The human and mouse ZRS sequence is highly conserved. The human ZRS has 19 putative ETS-binding sites, all with affinities equal to or lower than 0.52. The mouse ZRS has 6 functionally validated and 12 putative ETS sites; 15 of these are conserved in location and affinity with the human ZRS. **b**, The ETS-A site and surrounding sequence show perfect conservation between mouse and human, as indicated by asterisks. Blue box highlights the ETS-A sequence within the human and mouse ZRS.

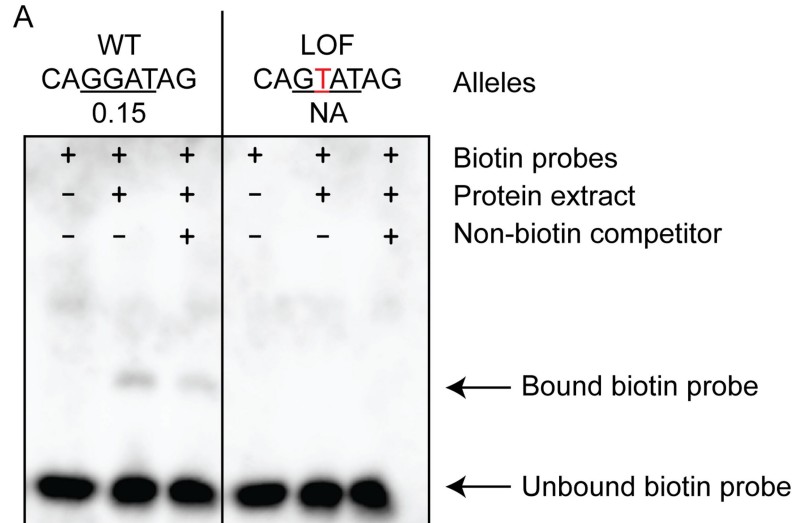

A

WT
CAGGATAG
0.15

LOF
CAGTATAG
NA

Alleles

| + | + | + | + | + | + | Biotin probes |
| − | + | + | − | + | + | Protein extract |
| − | − | + | − | − | + | Non-biotin competitor |

← Bound biotin probe

← Unbound biotin probe

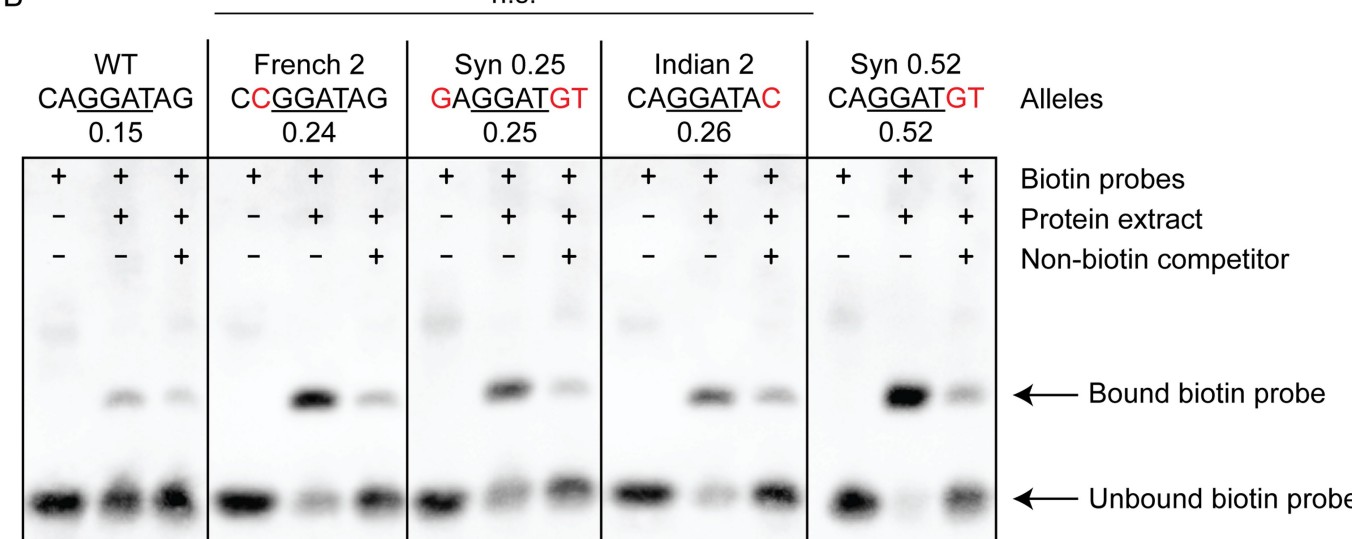

B

n.s.

WT
CAGGATAG
0.15

French 2
CCGGATAG
0.24

Syn 0.25
GAGGATGT
0.25

Indian 2
CAGGATAC
0.26

Syn 0.52
CAGGATGT
0.52

Alleles

Biotin probes
Protein extract
Non-biotin competitor

← Bound biotin probe

← Unbound biotin probe

**Extended Data Fig. 4 | EMSA shows the binding of human and mouse ETS-1 to the ETS-A site and ETS-A variants. a,** WT ETS-A probe sequence can bind to the ETS-1 DBD to generate a band, the intensity of which decreases as the biotin-labelled probe is outcompeted by a non-biotin competitor probe. A single-bp change in the ETS-A site (LOF) leads to a loss of binding signal. **b,** EMSAs for WT, French 2, Syn 0.25, Indian 2 and Syn 0.52 sequences. The total amount of bound probe relative to the unbound probe is not statistically different between French 2, Indian 2 and Syn 0.25 sequences, suggesting that all three sequences have the same affinity. $P = 0.18$, one-way ANOVA. The Syn 0.52 sequence binds more strongly than the 0.25 or 0.15 sequences do. EMSAs were performed independently twice and both replicates show similar results. For gel source data, see Supplementary Fig. 1.

## Hindlimb

## Forelimb

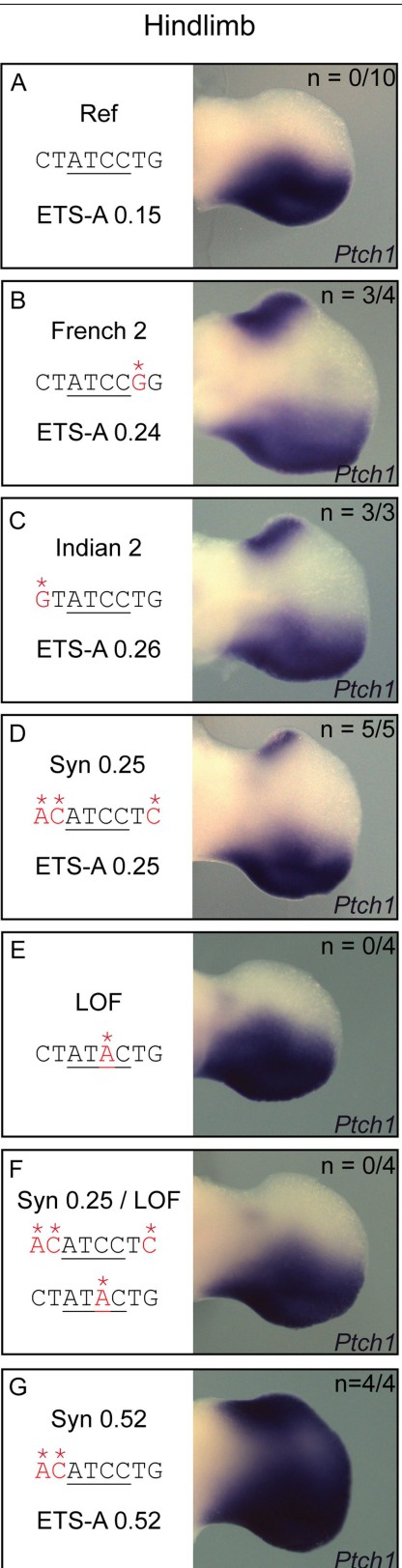

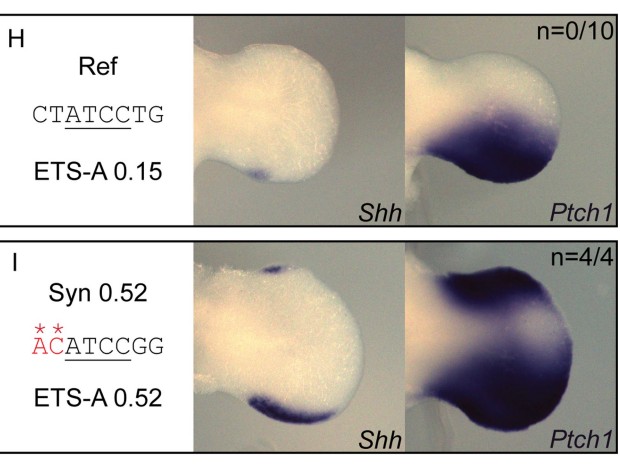

A  Ref  CTATCCTG  ETS-A 0.15  n = 0/10  *Ptch1*

B  French 2  CTATCCGG  ETS-A 0.24  n = 3/4  *Ptch1*

C  Indian 2  GTATCCTG  ETS-A 0.26  n = 3/3  *Ptch1*

D  Syn 0.25  ACATCCTC  ETS-A 0.25  n = 5/5  *Ptch1*

E  LOF  CTATACTG  n = 0/4  *Ptch1*

F  Syn 0.25 / LOF  ACATCCTC  CTATACTG  n = 0/4  *Ptch1*

G  Syn 0.52  ACATCCTG  ETS-A 0.52  n=4/4  *Ptch1*

H  Ref  CTATCCTG  ETS-A 0.15  n=0/10  *Shh*  *Ptch1*

I  Syn 0.52  ACATCCGG  ETS-A 0.52  n=4/4  *Shh*  *Ptch1*

**Extended Data Fig. 5 | *Ptch1* in situ hybridization in the hindlimb bud and forelimb bud of transgenic mice.** Embryos were collected at around the 53-somite stage for *Ptch1* in situ hybridization. **a**, *Ptch1* expression is restricted to the posterior domain in WT hindlimb buds. Ectopic *Ptch1* expression can be seen in the anterior domain of homozygous French 2 (**b**), Indian 2 (**c**), Syn 0.25 (**d**) hindlimb buds. LOF homozygotes (**e**) and embryos with Syn0.25/LOF alleles (**f**) do not have ectopic *Ptch1* expression as predicted. **g**, Syn 0.52 hindlimb

buds have larger domain of ectopic *Ptch1* expression than embryos with approximately 0.25 affinity ETS-A sites. **h**, In WT forelimb buds, *Shh* and *Ptch1* expression is restricted to the posterior domain. **i**, Syn 0.52 homozygotes display ectopic *Shh* expression and *Ptch1* expression in the forelimb buds. The *Shh* in situ hybridizations shown were performed on embryos at around the 48-somite stage. All limb bud images were acquired and cropped using the same settings.

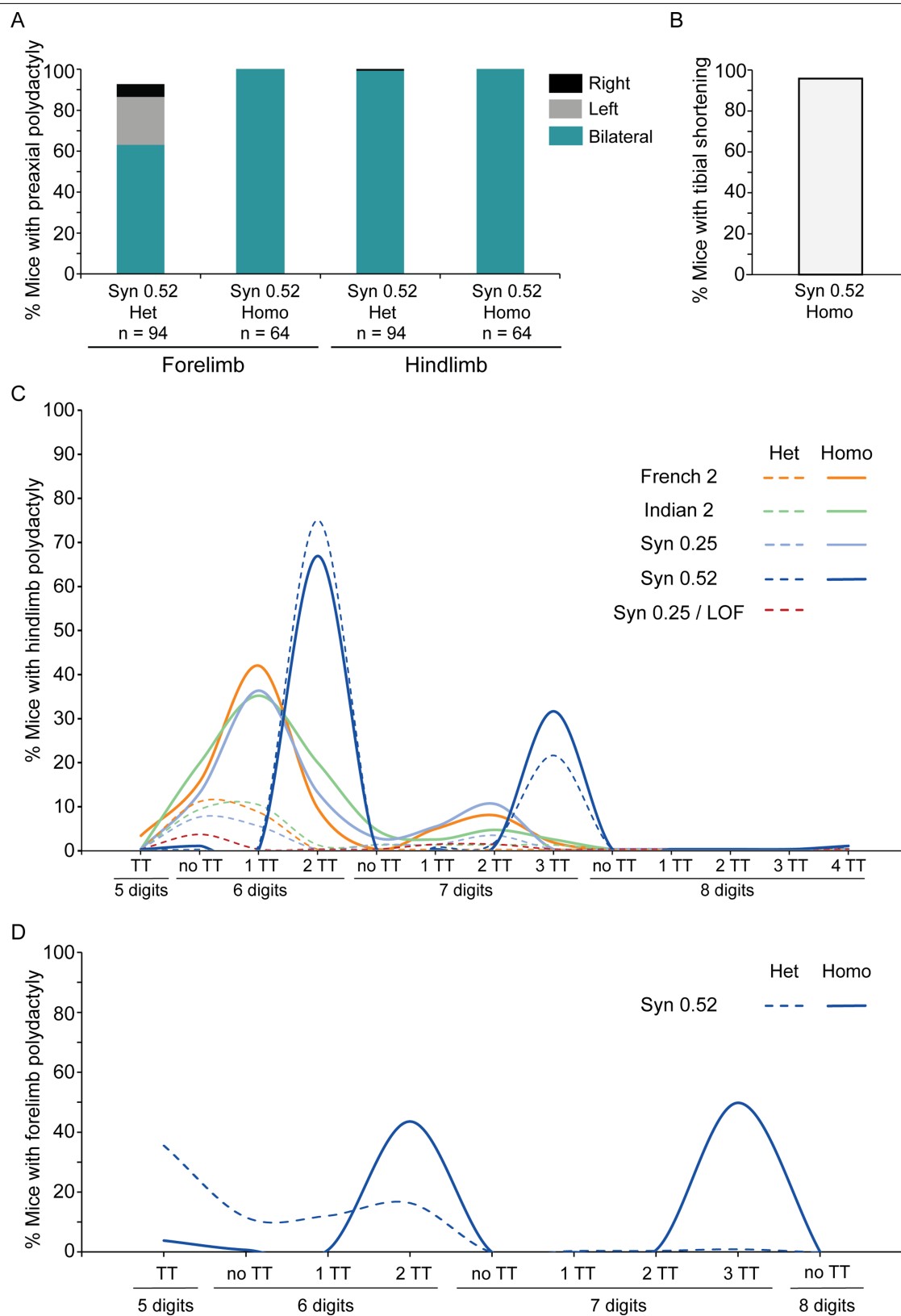

**Extended Data Fig. 6 | Syn 0.52 mice show highly penetrant polydactyly in both forelimb and hindlimb, as well as tibial hemimelia. a**, In the hindlimb, heterozygotes and homozygotes have 100% penetrance in polydactyly. In the forelimb, homozygotes have 100% penetrance whereas heterozygotes have 93.6% penetrance. **b**, Tibial hemimelia is observed in 95.3% of homozygotes but no heterozygotes. **c**, Digit phenotypes on the hindlimbs of all mice studied. **d**, Digit phenotypes on the forelimbs of Syn 0.52 mice. WT, LOF, French 2, Indian 2 and Syn 0.25 have no forelimb phenotypes. TT denotes triphalangeal toe or thumb.

**Extended Data Fig. 7 | Affinity-optimizing SNVs are significantly associated with GOF enhancer activity.** Enhancers containing SNVs that increase ETS affinity are significantly enriched in GOF expression relative to all other SNVs within a 2% sequence mutagenesis assay of the ZRS enhancer[5]. The bounds of the box plots define the 25th, 50th and 75th percentiles, and whiskers are 1.5 × the interquartile range. One-tailed Mann–Whitney *U* test.

A    <u>HOXA13</u>

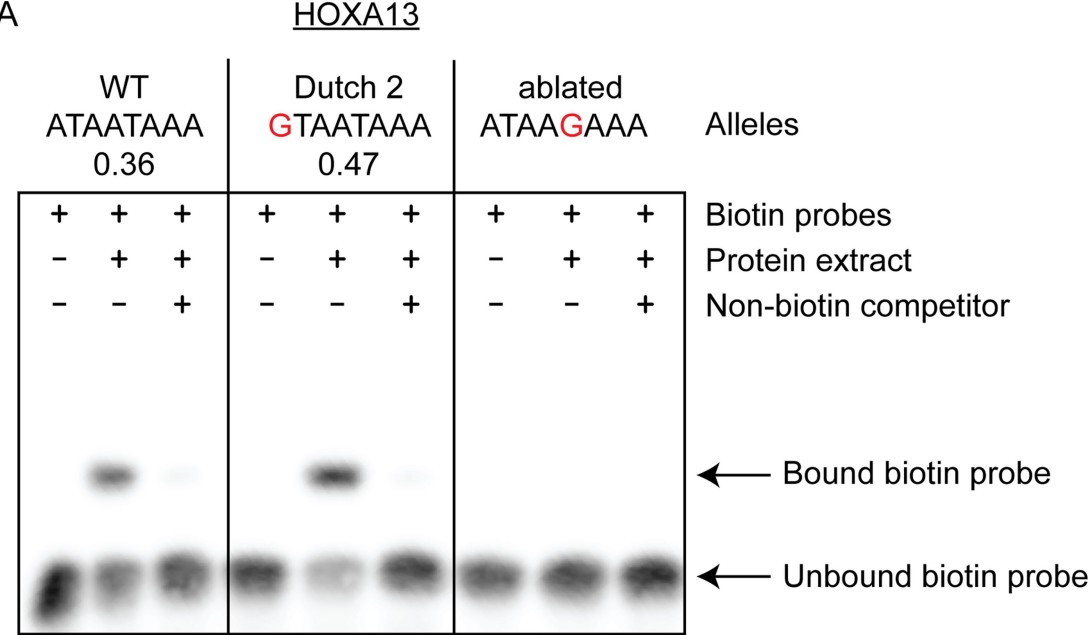

B    <u>HOXD13</u>

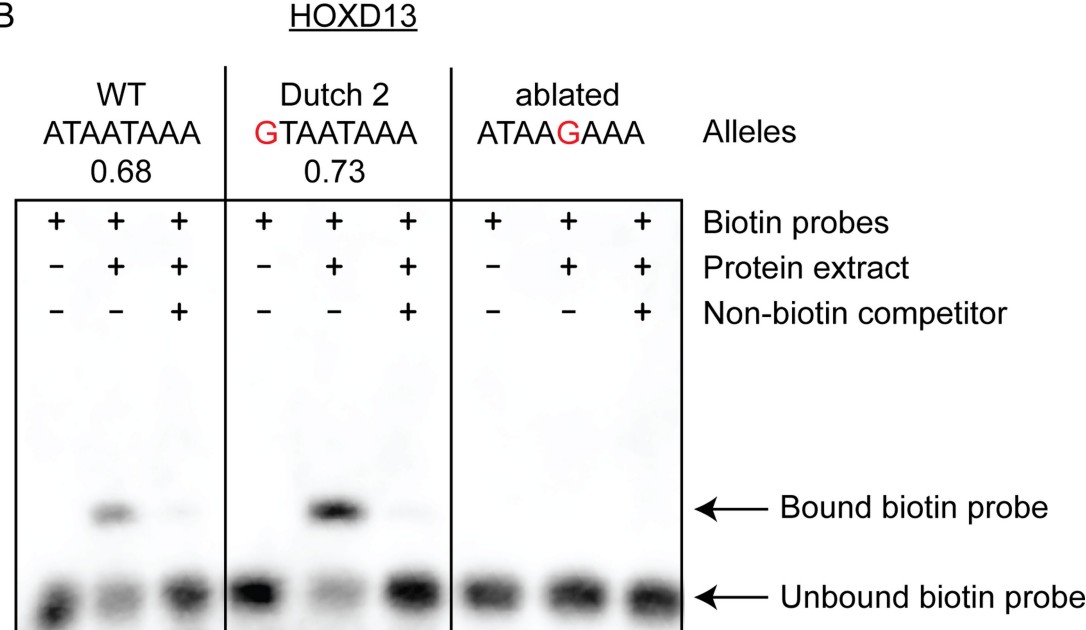

**Extended Data Fig. 8 | EMSA shows stronger binding of the human HOXA13 and HOXD13 DBDs to the Dutch 2 variant relative to the WT sequence. a**, WT probe sequence can bind to the human HOXA13 DBD to generate a band, the intensity of which decreases in the presence of the non-biotin competitor probe. The Dutch 2 variant binds to HOXA13 more strongly than does the WT allele. A single-bp change that ablates HOXA13 binding shows no binding. **b**, EMSA with the same WT and Dutch 2 probe sequences as in **a**, performed with the human HOXD13 DBD. The Dutch 2 variant binds to HOXD13 more strongly than does the WT allele. EMSAs were performed independently twice and both replicates show similar results. For gel source data, see Supplementary Fig. 1.

| Enhancer | Disease associated | Cell line | |
|---|---|---|---|
| BCL11A | Sickle cell disease | HEL92.1.7 | Erythroblast cell line |
| IRF4 | Human pigmentation | SK-MEL-28 | Melanoma cell line |
| IRF6 | Cleft lip | HaCat | Keratinocyte cell line |
| MYC (rs6983267) | Various types of cancer (SNP is associated with colorectal cancer development) | HEK293T + 20mM LiCl | Derived embryonic kidney cell line (activated Wnt pathway) |
| MYC (rs11986220) | Various types of cancer (SNP is associated with prostate cancer) | LNCaP + 100nM DHT | Prostate adenocarcinoma cell line (stimulated androgen activity) |
| RET | Hirschsprung's disease | Neuro-2a | Mouse neuroblastoma cell line |
| SORT1 | Plasma low-density lipoprotein cholesterol & myocardial infarction | HepG2 | Hepatocellular carcinoma cell line |
| TCF7L2 | Type 2 diabetes | MIN6 | Mouse pancreatic beta cell line |
| UC88 | (ultraconserved enhancer with strong forebrain activity) | Neuro-2a | Mouse neuroblastoma cell line |
| ZFAND3 | Type 2 diabetes | MIN6 | Mouse pancreatic beta cell line |
| ZRS | Polydactyly and other limb malformations | NIH/3T3 (+ HOXD13 + HAND2) | Mouse fibroblast cells transfected with limb bud TFs to give a limb-like cell identity |

**Extended Data Fig. 9 | Experimental details for MPRA performed with 11 disease-associated enhancers.** Eleven disease-associated enhancers tested in saturation mutagenesis MPRAs; table modified from a previous study[30]. Two different MYC enhancers were assayed. UC88 is an ultraconserved enhancer. The MPRA for each enhancer is performed within cell lines relevant to the disease studied, as detailed.

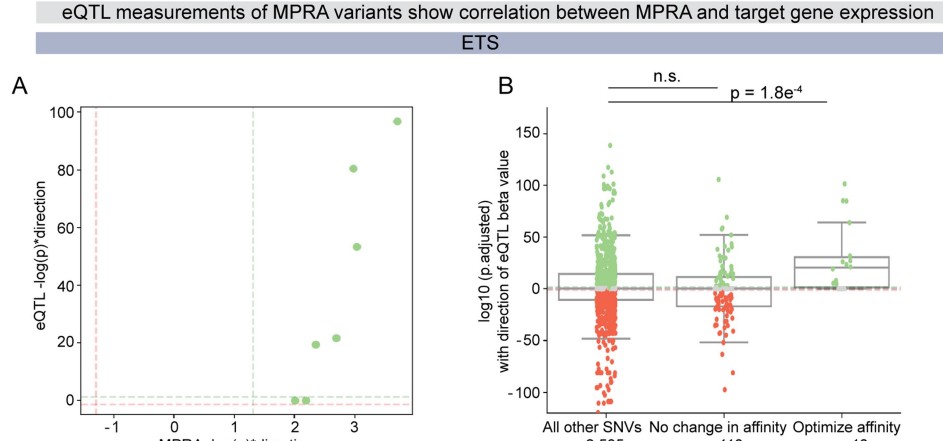

eQTL measurements of MPRA variants show correlation between MPRA and target gene expression

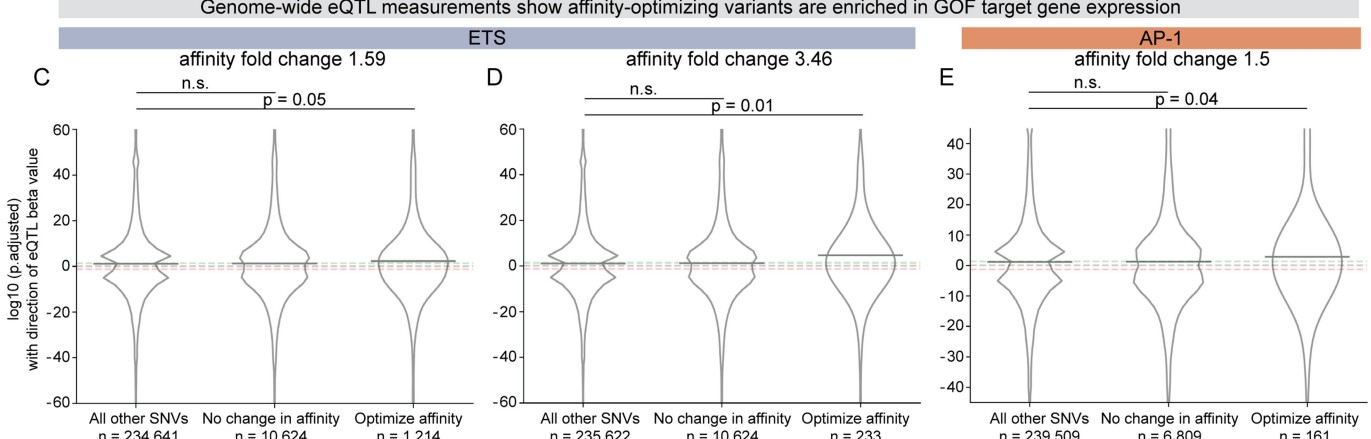

Genome-wide eQTL measurements show affinity-optimizing variants are enriched in GOF target gene expression

**Extended Data Fig. 10 | Affinity-optimizing eQTL variants are enriched in GOF target gene expression. a**, Seven ETS affinity-optimizing variants within lymphoblastoid regulatory element MPRA drive significant GOF reporter expression[40]. Five of these are associated with significant eQTL differential expression and all of these SNVs drive increased target gene expression[41]. Dotted lines indicate thresholds for significance at $P < 0.05$ (significant GOF in green, and significant LOF in red). **b**, eQTL analysis for all MPRA variants tested within the lymphoblastoid regulatory elements (regardless of variant effect within MPRA assay)[40,41]. ETS affinity-optimizing SNVs are enriched in GOF target gene

expression. The bounds of the box plots define the 25th, 50th and 75th percentiles, and whiskers are 1.5 × the interquartile range. One-tailed Mann–Whitney $U$ test. **c**–**e**, Genome-wide analyses for all significant eQTLs ($P < 0.01$) within the lymphoblastoid cell line[41] show that affinity-optimizing variants are enriched for GOF expression. Line indicates mean values. One-tailed Mann–Whitney $U$ Test. **c**, ETS SNVs with ≥1.59 affinity fold change. **d**, ETS SNVs with ≥3.46 affinity fold change; this is the fold change of 0.52 mice relative to WT 0.15 ETS-A. **e**, AP-1 SNVs ≥1.5 affinity fold change.

# Reporting Summary

## Statistics

For all statistical analyses, confirm that the following items are present in the figure legend, table legend, main text, or Methods section.

| n/a | Confirmed | |
|---|---|---|
| ☐ | ☒ | The exact sample size (*n*) for each experimental group/condition, given as a discrete number and unit of measurement |
| ☐ | ☒ | A statement on whether measurements were taken from distinct samples or whether the same sample was measured repeatedly |
| ☐ | ☒ | The statistical test(s) used AND whether they are one- or two-sided<br>*Only common tests should be described solely by name; describe more complex techniques in the Methods section.* |
| ☒ | ☐ | A description of all covariates tested |
| ☐ | ☒ | A description of any assumptions or corrections, such as tests of normality and adjustment for multiple comparisons |
| ☐ | ☒ | A full description of the statistical parameters including central tendency (e.g. means) or other basic estimates (e.g. regression coefficient) AND variation (e.g. standard deviation) or associated estimates of uncertainty (e.g. confidence intervals) |
| ☐ | ☒ | For null hypothesis testing, the test statistic (e.g. *F*, *t*, *r*) with confidence intervals, effect sizes, degrees of freedom and *P* value noted<br>*Give P values as exact values whenever suitable.* |
| ☒ | ☐ | For Bayesian analysis, information on the choice of priors and Markov chain Monte Carlo settings |
| ☒ | ☐ | For hierarchical and complex designs, identification of the appropriate level for tests and full reporting of outcomes |
| ☒ | ☐ | Estimates of effect sizes (e.g. Cohen's *d*, Pearson's *r*), indicating how they were calculated |

*Our web collection on statistics for biologists contains articles on many of the points above.*

## Software and code

Policy information about availability of computer code

| Data collection | No custom code was used in data collection. |
|---|---|
| Data analysis | Standard python libraries were used to analyze MPRA, ChIP and eQTL data (pandas 1.4.2 , numpy 1.22.3, statsmodels 0.13.2, seaborn 0.11.2 , bx 0.9.0, pyliftover 0.4, Bio 1.79, scipy, matplotlib). Custom python code utilizing these libraries were used to process, visualize and do statistics. Python version 3.9.7 was used. ImageLab Software Version 6.1 was used for EMSA analysis. |

For manuscripts utilizing custom algorithms or software that are central to the research but not yet described in published literature, software must be made available to editors and reviewers. We strongly encourage code deposition in a community repository (e.g. GitHub). See the Nature Portfolio guidelines for submitting code & software for further information.

## Data

Policy information about availability of data

All manuscripts must include a data availability statement. This statement should provide the following information, where applicable:

- Accession codes, unique identifiers, or web links for publicly available datasets
- A description of any restrictions on data availability
- For clinical datasets or third party data, please ensure that the statement adheres to our policy

All data supporting the findings of this study are available within the paper and its Supplementary Information. ChIP-Seq data analyzed in this paper were downloaded using accession codes GSM2218592, GSM3520734, GSM4110116. eQTL data analyzed in this paper was downloaded from the EBI eQTL catalogue

# Research involving human participants, their data, or biological material

Policy information about studies with human participants or human data. See also policy information about sex, gender (identity/presentation), and sexual orientation and race, ethnicity and racism.

| | |
|---|---|
| Reporting on sex and gender | Not applicable |
| Reporting on race, ethnicity, or other socially relevant groupings | Not applicable |
| Population characteristics | Not applicable |
| Recruitment | Not applicable |
| Ethics oversight | Not applicable |

Note that full information on the approval of the study protocol must also be provided in the manuscript.

# Field-specific reporting

Please select the one below that is the best fit for your research. If you are not sure, read the appropriate sections before making your selection.

☒ Life sciences  ☐ Behavioural & social sciences  ☐ Ecological, evolutionary & environmental sciences

For a reference copy of the document with all sections, see nature.com/documents/nr-reporting-summary-flat.pdf

# Life sciences study design

All studies must disclose on these points even when the disclosure is negative.

| | |
|---|---|
| Sample size | We sought to study as many mice as possible, we stopped when we had around 80 mice for each genotype for heterozygotes and close to 40 for homozygotes. |
| Data exclusions | We have not excluded any data. |
| Replication | We analyzed mice from multiple litters and repeated all experiments at least twice with similar results. |
| Randomization | We did not need to do this for our experimental design, since there are no applicable covariates to randomize for in this study. |
| Blinding | Each mouse born into our colony has all 4 limbs inspected by an investigator blind to genotype at postnatal day 10-18 during routine ear clipping (for identification) and tail biopsy collection (for genotyping). |

# Reporting for specific materials, systems and methods

We require information from authors about some types of materials, experimental systems and methods used in many studies. Here, indicate whether each material, system or method listed is relevant to your study. If you are not sure if a list item applies to your research, read the appropriate section before selecting a response.

## Materials & experimental systems

| n/a | Involved in the study |
|---|---|
| ☐ | ☒ Antibodies |
| ☒ | ☐ Eukaryotic cell lines |
| ☒ | ☐ Palaeontology and archaeology |
| ☐ | ☒ Animals and other organisms |
| ☒ | ☐ Clinical data |
| ☒ | ☐ Dual use research of concern |
| ☒ | ☐ Plants |

## Methods

| n/a | Involved in the study |
|---|---|
| ☒ | ☐ ChIP-seq |
| ☒ | ☐ Flow cytometry |
| ☒ | ☐ MRI-based neuroimaging |

## Antibodies

| Antibodies used | 1:2500 anti-digoxigenin-AP antibody (Roche, 11093274910) |
|---|---|
| Validation | Other papers that have also used this antibody and found it effective: PMID33412105, PMID36184733, PMID30945286 |

## Animals and other research organisms

Policy information about <u>studies involving animals</u>; <u>ARRIVE guidelines</u> recommended for reporting animal research, and <u>Sex and Gender in Research</u>

| Laboratory animals | Mice C57Bl/6NHsd, mice were phenotyped at 10-18 days old. Mice are housed at room temperature (between 68-79°F) and regular humidity. Since housing temperature and humidity are ambient and not relevant to the study, we did not include this information in the manuscript. All other relevant housing conditions for mice is included in the methods section. |
|---|---|
| Wild animals | The study did not involve wild animals. |
| Reporting on sex | Mice of both sexes were involved in this study. Sex is determined during 10-21 days old by comparison of urogenital-anal distance and presence of female nipples or male genitalia. Phenotypes of mice based on sex is reported in Supplementary Materials. |
| Field-collected samples | This study did not involve samples collected from the field. |
| Ethics oversight | All animal procedures and studies were approved by the University of California, San Diego Institutional Animal Care and Use Committee according to the Association for Assessment and Accreditation of Laboratory Animal Care guidelines. |

Note that full information on the approval of the study protocol must also be provided in the manuscript.

