## [Peer Review File · Nature]

Manuscript Title: Affinity-optimizing enhancer variants disrupt development

Reviewer Comments & Author Rebuttals

Reviewer Reports on the Initial Version:

Referees' comments:

Referee #1 (Remarks to the Author):

The manuscript by Lim & Ryan et al investigates the role of enhancers in controlling gene expression and how variants within enhancers can cause diseases. The ZRS enhancer mediates expression of Shh and is critical for limb and digit formation. Despite decades of study, the mechanisms by which enhancer variants alter phenotypes are poorly understood. The authors now present new findings that show low-affinity ETS sites within the ZRS enhancer and how small changes in the affinity of these sites caused by SNVs can lead to polydactyly with different severities. The ZRS is by far the best studied enhancer with over 30 SNVs known to cause limb phenotype. The authors show that two famous mutations in the ZRS called 'French 2' and 'Indian 2' which had been studied in LacZ report essays before give the same phenotypes in mice and increase the affinity of ETS-A by the same amount. They then show in several mouse models that the mechanism driving polydactyly in the human mutations are the very subtle increase in affinity of the ETS-A site. Finally, they show the ability to predict the relationship between genotype and phenotype for SNVs within the ETS-A site and validated their findings in over 700 mice. The authors then aim to generalize their findings to other transcription factors and enhancers by investigating several MPRA datasets. They finally conclude that searching for affinity optimizing SNVs could be an effective and simple way to prioritize causal enhancer variants for further functional analysis.

This beautiful manuscript addresses one of the central questions of human genetics of the next decade: the influence of variation in the non-coding genome as the cause of human disease. Furthermore, the paper offers an 'effective and simple' method to predict genotype-phenotype collations.

I am really excited about the manuscript and think it could be extremely relevant for the broad readership of Nature.

However, I have several concerns with approach and the manuscript in its current format

Major comment:

I have only one major issues with the manuscript in its present form mainly related to the approach that was taken by the authors.

By focusing on an extremely well-studied example namely the ZRS and GOF mutations the authors draw some impressive conclusions. However, it needs to be kept in mind that the ZRS is an extremely unusual enhancer in terms of sequence conservation and the fact that SNVs have such dramatic phenotypic effects. Recent data indicate that most enhancers seem to behave very differently than the ZRS and are very robust against any kind of sequence change. Enhancer redundancy and shadow enhancers provide important mechanisms for buffering gene expression against mutations in non-coding regulatory regions of genes implicated in human disease (Nature volume 554, pages 239–243 2018; Nature Reviews Genetics volume 22, pages 324–336 2021). In particular the Osterwald paper (Nature 2018) showed that none of the deletions of 10 individual limb enhancers actually showed any noticeable changes in limb morphology in mice. At the same time none of the recent large sequencing projects have been able to find a substantial number of non-coding variants in enhancers elements as the cause human disease (N Engl J Med 2021 Nov

11;385(20):1868-1880; & Genet Med. 2022 Nov;24(11):2296-2307.)

In short, the ZRS seems to be an exception and some conclusions drawn from it might be biased or misleading.

All this does not mean that the data presented in this paper are not exciting and extremely relevant. The finding alone that subtle increases in binding in ETS-A affinity can cause developmental defects and that larger affinity increases lead to more severe and penetrant phenotypes would be worth publishing.

However, my question would be if the authors can validate their findings on a different enhancer/locus in vitro or in vivo?

While I realize that the in silico analysis on the MPRA data look promising I think an independent validation would really strengthen the story of the manuscript. In particular, since the ZRS MPRA in cell lines is extremely artificial and did not even pick up the signal of the French 2 and Indian 2 variants. Another weakness of the MPRA approach is that the expression of the actual target gene is not measured. So GoF refers in the case of MPRA only to expression of the reporter construct itself (Kircher NCOMM 2020). Therefore, a secondary validation of the target gene is important here.

The question would then be, how does the affinity-optimizing SNV model relate to any of the 10 enhancer tested in the Osterwald paper (Nature 2018)? Could you force a gain function at these loci (for examples of Gli3)? Or the FTO locus, which is another famous example of a gain of function variant?

Minor comments:

1. How does the concept of increased ETS-A binding affinity relate to the ZRS duplication phenotypes Haas-type polysyndactyly and Laurin-Sandrow syndrome (J Med Genet. 2008 Sep;45(9):589-95 & J Med Genet. 2008 Jun;45(6):370-5)? The fact that smaller duplications are associated with a stronger polydactyly phenotype seems interesting in this context (Clin Genet 2014 Oct;86(4):318-25).

2. Figure 1:

- Could the authors include all known ZRS mutations in Panel A. This would be particularly interesting in relation to the ETS sites.

- Figure 1 in its current format doesn't seem like a standalone figure and could be combined with figure 2 to a multipanel figure.

3. Also figure 4 and 5 could be combined to one multipanel figure

4. In the abstract and in the discussion the MPRA results are being presented in misleading way.

Abstract: '...and within a wide variety of other disease-associated enhancers also lead to gain-of-function(GOF) gene expression'.

Discussion: 'Indeed we see that ETS affinity-optimizing variants are significantly enriched in GOF gene expression in two independent MPRA assays across many enhancers implicated in disease.'

The papers cited here Kircher et al. and Tewhey et al. do not measure gene expression of the target genes but simply measure the expression of the reporter construct. This is a big difference and needs to be discussed. See also my comment above.

Referee #2 (Remarks to the Author):

The study by Lim et al. is very interesting and original as it shows that even slight changes in the low affinity binding site of the transcription factors ETS cause gain-of-function congenital limb

malformations. The core of the study deals with the analysis of affinity-gain single nucleotide variants (SNV) in the ETS binding sites within the distant enhancer (ZRS) that regulates Shh expression during limb bud development. Previous studies had identified several such SNVs in the ZRS and established a causal link between ectopic Shh activation and anterior polydactyly in different humans and different other species. However the underlying molecular mechanism remained unknown. This study identifies the underlying primary cause, namely that these and other SNV increases the affinity of the ETS interaction with the mutant binding sites. The study provides compelling molecular and in vivo evidence that even small affinity increases cause anterior ectopic Shh expression with associated preaxial digit polydactyly. In addition, the authors re-analyze published datasets by others to provide evidence that SNV-mediated increase of binding affinities seem a more general cause underlying gain-of-function (GOF) pathologies, which provide good evidence for the predictive value of SNV-associated affinity increase with respect to pathologies. This is an exciting and largely unexpected advance that is of direct relevance to the understanding of pathologies caused by SNV in non-coding regions and/or known enhancers. Therefore, the study is of great general relevance and interest.

Points to be addressed in revision:

1. French 2 and Indian ZRS SNV analysis in transgenic mice: as there is no digit phenotypes in forelimb buds, it is important to know if Shh expression is unaltered, i.e. no anterior ectopic expression is detected both at early and late stages of forelimb bud development. If yes, this could be due to the fact that mouse fore- and hindlimb development are heterochronic, with the hindlimb bud being delayed by about half a developmental day? Or what are alternate explanations for this?. Please comment in the result section.
2. Related to this, the anterior ectopic digits are only detected in hindlimb buds, the description of the phenotype should reflect this: "The additional anterior digit in mouse hindlimb buds bears resemblance to the extra triphalangeal thumb observed in the orthologous human congenital malformations". This reviewer is of the opinion that calling it an extra triphalangeal thumb is not accurate as this phenotype is hindlimb specific in mouse.
3. Figure 3 (panels E-G): in comparison to the GOF SNV limb bud, development of the LOF SNV limb bud seems delayed in comparison to the GOF SNV limb bud. This needs to be clarified by accurate somite staging and/or including a comparative developmental time course as the LOF mutation is used to generate trans-heterozygous mice subsequently (e.g. Figure 4B).
4. These results are unexpected or spectacular, but not "shocking". This is the wrong term used several times in the manuscript text. For example previous studies have established that low levels of anterior-ectopic Shh expression are sufficient to cause preaxial polydactyly digit phenotypes in mice. It is amazing that SNVs cause affinity increases that result in anterior ectopic Shh expression. Can the authors please discuss /speculate why such a small affinity increase could result in anterior ectopic Shh activation rather than a posterior increase.
5. From the section entitled "Affinity-optimizing ETS SNVs across the ZRS drive GOF gene expression", the description of the results becomes much more difficult to follow. This reviewer had to repeatedly check the 3 original manuscripts describing the datasets used to understand what was done. This criticism applies to the last 3 sections of the results. This reviewer is of the opinion, that the authors have to introduce/summarize these published studies much better before they describe their analysis. Also, all abbreviations taken over from the original studies must be clearly explained. This final part of the result section is very important but its description has to be improved such that it can stand alone, i.e. without readers having to read the other studies beforehand.
6. Related to this: the published datasets did not identify the French 2 and Indian 2 SNV as pathological variants with increased affinity for ETS binding, a highlight of the first part of the manuscript. To what extent does this alter the value and predictive power of the datasets used for analysis?
7. The robustness of GOF phenotypes is impressive in inbred C57Bl6 mice. Is there any published evidence for such robustness in different humans carrying the same SNVs? It would be great if the authors could comment on this.

Minor points:

8. Extended Data Figure 4 panel B. The legend is somewhat confusing and should be better described. This reviewer understands the sentence in question as follows: The total amount of biotinylated probe (bound and unbound fractions) is not statically different between the different samples. Please clarify.

9. In the Excel file, the Supplementary Tables are not labelled.

Referee #3 (Remarks to the Author):

Lim and colleagues address an important problem in gene regulation, how enhancers regulate genes and how single nucleotide variants (SNVs) can change their activity. As eluded in the intro, enhancers serve as transcription factor (TF) binding platforms that translate protein-DNA interaction into transcriptional outputs. However, how the sequence composition defines activity and how mutations will alter its function remains largely unknown. The authors decided to study this important question in one of the most well studied enhancers, the ZRS, the sole limb enhancer of *Shh*. Mutations in the ZRS are known to alter *SHH* activity in the limb thereby leading to increased and ectopic expression which in turn results in limb malformation. The activity of this enhancer has been used to study enhancer activity and gene regulation also because it is the only regulator of *Shh* in the limb, which appears to be rather an exception than the rule in enhancer-driven gene regulation.

The authors find that the known TF binding sites for the TF ETS have very low binding affinities as measured by Protein Binding Microarray (PBM). They also provide convincing data that this measurement reflects reality *in vitro* and *in vivo*. They have the hypothesis that mutations that increase TF binding drive pathogenic gain of enhancer function. Indeed, they can show that previously published mutations increase binding affinity by a small degree and this results in a) ectopic *Shh* expression and b) a limb phenotype. Furthermore, the authors produce a mutation that is predicted to increase affinity to the same degree as the other mutations and show that this change has very similar effects. In a further assay they show that an increase in binding affinity goes along with a more severe phenotype, also measured convincingly *in vivo*. The authors go on to test if this principle holds true in other experimental settings and other TFs. Again, they show that a gain of function in binding goes along with an increase in expression. They show that this mechanism can be used to predict the effect of SNVs on enhancer activity and pathogenicity. Overall, this is a well written and carefully conducted study of high relevance. One of the major problems in current genetics is our lack of ability to predict the outcome of SNVs on gene expression and disease. This ms shows that very small changes can have major effects. More importantly, it shows that current models to predict biological relevant TF bindings sites may not be correct. If low affinity bindings sites are generally the more important ones, one would have to re-design the general analysis strategy of non-coding SNVs.

The strength of this paper is the *in vitro* analysis in combination with the careful and impressive mouse genetics together with the comprehensive follow up phenotyping analysis. Such an *in vivo* analysis is essential to the field which to a large extent relies on *in vitro* episomal assay. Overall, the results are highly convincing and robustly demonstrate how single sequence variants can disrupt the careful balance of TF binding at enhancers to cause disease.

On the other side, the ZRS has been studied extensively and that suboptimal TF binding defines enhancer activity has been demonstrated previously. Furthermore, *Shh* is a rare example in which the expression domain of a gene is driven by a single enhancer. The situation may be very different in other cases where multiple enhancers drive complex expression patterns, a fact that is not discussed. Nevertheless, the authors provide additional information that allows for this type of generalization. In fact, their demonstration that other TFs such as AP1 may function in a similar manner gives important insights beyond the ZRS example.

In summary, this is a study with carefully designed *in vitro* and *in vivo* experiments that provides convincing evidence that low affinity bindings sites are important for enhancer function and that

gain of function is a likely general mechanism how SNVs can cause disease.

Major comments (in no particular order)

1. There are many redundancies in the present version of the paper. Some of the intro information is repeated in the abstract, the intro and the discussion.
2. The discussion is too long and has many redundant parts. For example, the reader does not need another introduction about variants and enhancers.
3. The authors like the term "shockingly" – not appropriate for this type of ms.
4. The authors claim that they can predict outcome based on sequence and the prediction of binding affinity (Fig. 5). Many mutations that reside within the ZRS have been described in the literature. They are associated with phenotypes of different severity (see OMIM *605522). For example, some mutations result in polydactyly only, or in a much more severe phenotype involving the lower limb similar to the mutation in Fig. 5 (e.g. vander Meer et al. 2014). If their prediction is valid, it should also work when correlating human mutations with phenotypes.
5. How do the other so far reported mutations fit into the concept? The authors should show them and predict their binding affinity. Do they all result in and increase?
6. Duplications of the entire ZRS can also result in similar phenotypes (e.g. Wu et al. 2009). How do the authors explain this effect?
7. What is the model of the authors how an increase in TF binding can result in ectopic expression?

Reviewer #1:

The manuscript by Lim & Ryan et al investigates the role of enhancers in controlling gene expression and how variants within enhancers can cause diseases. The ZRS enhancer mediates expression of *Shh* and is critical for limb and digit formation. Despite decades of study, the mechanisms by which enhancer variants alter phenotypes are poorly understood. The authors now present new findings that show low-affinity ETS sites within the ZRS enhancer and how small changes in the affinity of these sites caused by SNVs can lead to polydactyly with different severities. The ZRS is by far the best studied enhancer with over 30 SNVs known to cause limb phenotype. The authors show that two famous mutations in the ZRS called 'French 2' and 'Indian 2' which had been studied in LacZ report essays before give the same phenotypes in mice and increase the affinity of ETS-A by the same amount. They then show in several mouse models that the mechanism driving polydactyly in the human mutations are the very subtle increase in affinity of the ETS-A site. Finally, they show the ability to predict the relationship between genotype and phenotype for SNVs within the ETS-A site and validated their findings in over 700 mice. The authors then aim to generalize their findings to other transcription factors and enhancers by investigating several MPRA datasets. They finally conclude that searching for affinity-optimizing SNVs could be an effective and simple way to prioritize causal enhancer variants for further functional analysis.

This beautiful manuscript addresses one of the central questions of human genetics of the next decade: the influence of variation in the non-coding genome as the cause of human disease. Furthermore, the paper offers an 'effective and simple' method to predict genotype-phenotype collations. I am really excited about the manuscript and think it could be extremely relevant for the broad readership of Nature. However, I have several concerns with approach and the manuscript in its current format.

Major comment:

I have only one major issues with the manuscript in its present form mainly related to the approach that was taken by the authors. By focusing on an extremely well-studied example namely the ZRS and GOF mutations the authors draw some impressive conclusions. However, it needs to be kept in mind that the ZRS is an extremely unusual enhancer in terms of sequence conservation and the fact that SNVs have such dramatic phenotypic effects. Recent data indicate that most enhancers seem to behave very differently than the ZRS and are very robust against any kind of sequence change. Enhancer redundancy and shadow enhancers provide important mechanisms for buffering gene expression against mutations in non-coding regulatory regions of genes implicated in human disease (Nature volume 554, pages 239–243 2018; Nature Reviews Genetics volume 22, pages 324–336 2021). In particular the Osterwald paper (Nature 2018) showed that none of the deletions of 10 individual limb enhancers actually showed any noticeable changes in limb morphology in mice. At the same time none of the recent large sequencing projects have been able to find a substantial number of non-coding variants in enhancers elements as the cause human disease (N Engl J Med 2021 Nov 11;385(20):1868-1880; & Genet Med. 2022 Nov;24(11):2296-2307.) In short, the ZRS seems to be an exception and some conclusions drawn from it might be biased or misleading.

All this does not mean that the data presented in this paper are not exiting and extremely relevant. The finding alone that subtle increases in binding in ETS-A affinity can cause developmental defects and that larger affinity increases lead to more severe and penetrant phenotypes would be worth publishing. However, my question would be if the authors can validate their findings on a different enhancer/ locus in vitro or in vivo?

We thank the reviewer for their comments and agree that the ZRS is a somewhat unusual enhancer. We also agree that very few studies have identified and validated the role of enhancer variants on phenotype. As the reviewer is bringing up several points here, we have broken this down to address each of them.

Firstly, enhancer redundancy: We agree that most enhancers are redundant, and this means that loss of an individual enhancer typically has no phenotype as beautifully illustrated by Osterwalder et al 2018¹. Lettice et al 2012 also illustrated redundancy of ETS sites within the ZRS². It is due to this redundancy that our manuscript focuses on studying variants that results in GOF enhancer activity and expression. We hypothesized that the role of affinity-optimizing SNVs on GOF gene expression is widespread regardless of the type of enhancer as this increase in expression, either levels, spatially or temporally is hard to buffer. We have now added additional data to address the issue of enhancer redundancy.

Firstly, to see if affinity-optimizing SNVs drive GOF expression in both redundant and nonredundant enhancers at the level of reporter assays we have looked at a well characterized redundant enhancer, the Interferon-beta enhanceosome which regulates the Interferon-beta gene³⁻⁶. Within MPRA data testing variants of the Interferon-beta enhanceosome we see that SNVs that increase the binding affinity of IRF drive GOF enhancer activity. Overexpression of IRF3 has been shown to cause increased expression of endogenous IFN-beta and these cells have an enhanced antiviral state that better restricts viral replication⁷. Therefore, we anticipate that the affinity-optimizing SNVs that lead to GOF expression may contribute to increased expression of IFN-beta and an enhanced antiviral state, although this remains to be tested.

We appreciate that this data is still only MPRA analysis and so does not measure cellular phenotypes. To further address this we have compared the impact of variants within MPRA datasets and the endogenous locus using eQTL data. The lymphoblastoid regulatory MPRA dataset that we used was constructed to test variants found in eQTL and determine which variants contribute to changes in gene expression⁸. We now compare the MPRA data to the impact on target gene expression as determined by eQTL analysis. Of the 7 ETS affinity-optimizing SNVs that drive GOF expression in the MPRA, 5 of these variants are significant eQTLs and all 5 of eQTLs are associated with an increase in expression of a target gene (Extended Data Figure 10). We did not do this analysis for AP-1 as there were only 3 MPRA variants found in the eQTL dataset. We next extended our analysis to look genome-wide. We find that eQTL affinity-optimizing SNVs show significant enrichment in increased expression of target genes (positive beta values), while SNVs that don't alter affinity of ETS sites show no enrichment in increased expression of target genes, and we see the same effect for AP-1. Furthermore, we find that higher fold change of ETS is associated with a more significant enrichment in GOF gene expression. These data suggest that affinity-optimizing SNVs drive GOF expression of target genes within the endogenous locus and that this GOF activity is not buffered within the endogenous context.

Second, many studies have not found causal variants within enhancers: We agree that many studies fail to identify causal enhancer variants within their datasets. In the examples referenced by the reviewer, if we are understanding the data correctly, casual variants were identified within 19-35% of the patients and all of these were in coding regions^{9,10}. However for the majority of patients no causal variant was identified, it is therefore possible that variants within regulatory elements could contribute to phenotypes in these remaining patients. In another study looking at intellectual disability the authors estimated that 1-3%

of patients could have a non-coding variant driving their neural developmental disorder¹⁰. Other studies suggest that at least 60% of variation associated with phenotypes lies within enhancers¹¹, and 93% of all GWAS variants are in non-coding regions¹². There is clearly a large disconnect between association studies suggesting a large number of variants contribute to phenotypes and functional data to demonstrate a causal link and much to be learned over the next few decades. As more studies link enhancer variants to phenotypes we'll find out the true contribution of enhancer variants to phenotypes.

However, my question would be if the authors can validate their findings on a different enhancer/locus in vitro or in vivo?

We have another manuscript that sees a similar phenomenon in the developing *Ciona* heart¹³. In this study, we focus on the FoxF enhancer which activates FoxF expression in the heart cells (known as the TVCS). Affinity-optimizing SNVs within FoxF ETS sites drive ectopic expression in the ATMs, which are not heart cells. This ectopic expression causes migration of non-heart progenitors (ATMs) to the ventral midline with the heart cells. Within 6% of animals this leads to heart defects including enlarged hearts and in the most extreme cases an extra beating heart. This study provides another example of how affinity-optimizing SNVs drive organismal level phenotypes. The manuscript can be found on bioRxiv¹³. We see this as a lower impact study to complement this manuscript as it focuses only on ETS and in the context of a marine invertebrate.

{REDACTED}

{REDACTED}

[REDACTED }

While I realize that the *in silico* analysis on the MPRA data look promising I think an independent validation would really strengthen the story of the manuscript. In particular, since the ZRS MPRA in cell lines is extremely artificial and did not even pick up the signal of the French 2 and Indian 2 variants.

We appreciate the reviewer's concern about the validity of the MPRA given its failure to detect GOF expression with French 2 and Indian 2 variants. These variants have very subtle and dynamic effects on gene expression, as we have demonstrated in our *in situ* hybridization. Even in a reporter assay that integrated these variant into the genome and

analyze expression in the developing limb bud, we see that only one of these variants is detected as driving expression¹⁹. Variants of larger predicted effects can be detected to drive significant changes in expression within MPRA datasets, for example the ZRS French 3 variant was detected in the MPRA and in the mouse endogenous locus in another publication¹⁹. While MPRA is a powerful tool for high-throughput analysis of enhancer variants to highlight true positives it lacks sensitivity to detect subtle or dynamic changes in gene expression. This is illustrated by the fact that these assays identifying variants such as French 3 that drive strong ectopic expression when tested in the endogenous locus but miss variants such as French 2 and Indian 2²⁰. As a result of this limitation, we expect that the MPRA analysis is missing some of the more subtle variants and so the MPRA analysis underestimates the number of affinity-optimizing SNVs that contribute to increased reporter expression. To strengthen the story we've added analysis of another saturation mutagenesis MPRA dataset for the Interferon-beta enhanceosome²¹. We find a significant enrichment of GOF reporter activity with affinity-optimizing SNVs. We have also found an orthogonal study that validates one of these affinity-optimizing SNVs drives GOF expression via a reporter assay²².

Another weakness of the MPRA approach is that the expression of the actual target gene is not measured. So GoF refers in the case of MPRA only to expression if the reporter construct itself (Kircher NCOMM 2020). Therefore, a secondary validation of the target gene is important here.

We agree that MPRA data has limitations as is not within the endogenous locus and agree that measuring target gene expression would be better. One of the MPRA datasets we analyzed in our manuscript tests top-associated eQTL variants in lymphoblastoid cell lines⁸. Thus for this dataset we can compare the directionality of expression seen in the MPRA with the direction of expression seen in the endogenous locus. We found 7 ETS affinity-optimizing variants that drive GOF expression in our lymphoblastoid regulatory element MPRA, five of these are eQTLs that are associated with expression changes, all 5 of these show GOF expression of the target gene within the eQTL analysis. We also see genome wide a similar effect for both ETS and AP-1.

Extended Data Figure 10. eQTL analysis shows a significant enrichment in affinity optimizing SNVs and GOF expression. (see paper for full legend)

The question would then be, how does the affinity-optimizing SNV model relate to any of the 10 enhancer tested in the Osterwalder paper (Nature 2018)? Could you force a gain function at these loci (for examples of Gli3)? Or the FTO locus, which is another famous example of a gain of function variant?

We'd love to be able to answer this experimentally. Within the 10 enhancer studied in the Osterwalder paper we anticipate that gain of expression of Shox2, FGF10, Tbx5, Sox9 and potentially Gli3 could lead to limb phenotypes²³⁻²⁸. Very little is known about the regulatory inputs to any of these enhancers and the effect of affinity-optimizing SNVs may not be fully penetrant. Therefore, to do these studies we would first need to confirm what factors regulate these enhancers, then make mouse lines along with controls, followed by extensively phenotyping of many mice to control for incomplete penetrance. These studies would take several years to complete, and we feel they are beyond the scope of our current manuscript but would make excellent follow-up studies.

We also agree that the FTO locus would be an ideal place to look for SNPs that relate to a phenotype. GWAS studies have identified SNPs within the FTO locus that are associated with increased risk of obesity^{29,30}. In the cerebellum, obesity-associated SNPs in this locus are associated with increased expression of IRX3, notably one of these SNPs is also associated with increased BMI³¹. However to our knowledge, no one has tested if these SNPs in combination or individually lead to GOF IRX3 expression when placed in the endogenous locus, nor has anyone validated if these SNPs lead to obesity in mice. To determine if the SNPs within this enhancer could be an affinity-optimizing SNV, we have looked for transcription factors that regulate the expression of IRX3 in the nervous system. The only factor we could find is NKX³². However, none of the known SNPs fall within NKX binding sites. We'd love to work on this in the future, however as none of the SNP in the FTO locus have been tested in the endogenous locus to causally link a SNP to ectopic IRX3 expression or obesity risk, and as we have very little understanding of the regulatory inputs into this enhancer we expect that this project would require several years to complete. We're excited about the prospect and will reach out to the labs that study this enhancer to see if we can collaborate in the future.

Minor comments:

1. How does the concept of increased ETS-A binding affinity relate to the ZRS duplication phenotypes Haas-type polysyndactyly and Laurin-Sandrow syndrome (J Med Genet. 2008 Sep;45(9):589-95 & J Med Genet. 2008 Jun;45(6):370-5)? The fact that smaller duplications are associated with a stronger polydactyly phenotype seems interesting in this context (Clin Genet 2014 Oct;86(4):318-25).

We agree that the fact that the smallest duplications cause the stronger polydactyly phenotypes is interesting. The ZRS duplication phenotypes in the aforementioned manuscripts involve duplication of large regions, the smallest region is 16kb long which encompass the entire ZRS³³. We do not know the mechanism by which the duplication drives polydactyly and why the smaller deletion drives the stronger phenotype, but we anticipate that the mechanisms by which this duplication drive polydactyly is much more complex than the single base pair changes we're observing. One possible mechanism by which the duplication drives a phenotype is that by having a second ZRS enhancer, the *Shh* promoter is activated at twice the frequency of the original enhancer, it is possible that larger duplications may contain other regulatory components such as CTCF sites that

interfere with enhancer promoter interactions. Further analysis of the duplications and the mechanisms by which they drive polydactyly would be beneficial to the community.

2. Figure 1:

- Could the authors include all known ZRS mutations in Panel A. This would be particularly interesting in relation to the ETS sites.

Thank you for the suggestion. We now include all known ZRS mutations in Panel A. We also include all known ZRS mutations in Supplementary Table 5 and annotate any known mechanisms for driving polydactyly. The list includes any mechanism identified either in previous publications or affinity-optimizations that we discovered in our study.

- Figure 1 in its current format doesn't seem like a standalone figure and could be combined with figure 2 to a multipanel figure.

Thank you for the suggestion, we have now combined Figure 1 and 2.

3. Also figure 4 and 5 could be combined to one multipanel figure

Thank you for the suggestion. While we agree that figures 4 and 5 (currently figures 3 and 4) could be combined, we have them separate to preserve the current flow in the text.

4. In the abstract and in the discussion the MPRA results are being presented in misleading way. Abstract: '...and within a wide variety of other disease-associated enhancers also lead to gain-of-function(GOF) gene expression.' Discussion: 'Indeed we see that ETS affinity-optimizing variants are significantly enriched in GOF gene expression in two independent MPRA assays across many enhancers implicated in disease.' The papers cited here Kircher et al. and Tewhey et al. do not measure gene expression of the target genes but simply measure the expression of the reporter construct. This is a big difference and needs to be discussed. See also my comment above.

We thank the reviewer for bringing to attention the potential confusion when discussing changes in reporter gene expression from MPRA analyses. Throughout the manuscript when referring to the MPRA we now emphasize that this is an MPRA assay and thus measuring reporter expression. We also add the eQTL analysis to look at target genes.

Reviewer #2:

The study by Lim et al. is very interesting and original as it shows that even slight changes in the low affinity binding site of the transcription factors ETS cause gain-of-function congenital limb malformations. The core of the study deals with the analysis of affinity-gain single nucleotide variants (SNV) in the ETS binding sites within the distant enhancer (ZRS) that regulates *Shh* expression during limb bud development. Previous studies had identified several such SNVs in the ZRS and established a causal link between ectopic *Shh* activation and anterior polydactyly in different humans and different other species. However the underlying molecular mechanism remained unknown. This study identifies the underlying primary cause, namely that these and other SNV increases the affinity of the ETS interaction with the mutant binding sites. The study provides compelling molecular and in vivo evidence that even small affinity increases cause anterior ectopic *Shh* expression with associated preaxial digit polydactyly. In addition, the authors re-analyze published datasets by others to provide evidence that SNV-mediated increase of binding affinities seem a more general cause underlying gain-of-function (GOF) pathologies, which provide good evidence for the predictive value of SNV-associated affinity increase with respect to pathologies. This is an exciting and largely unexpected advance that is of direct relevance to the understanding of pathologies caused by SNV in non-coding regions and/or known enhancers. Therefore, the study is of great general relevance and interest.

Points to be addressed in revision:

1. French 2 and Indian ZRS SNV analysis in transgenic mice: as there is no digit phenotypes in forelimb buds, it is important to know if *Shh* expression is unaltered, i.e. no anterior ectopic expression is detected both at early and late stages of forelimb bud development. If yes, this could be due to the fact that mouse fore- and hindlimb development are heterochronic, with the hindlimb bud being delayed by about half a developmental day? Or what are alternate explanations for this?. Please comment in the result section.

We see no ectopic expression of *Shh* in the forelimb at both early and late stages of forelimb development. We agree with the reviewer that the lack of phenotype in the forelimb is likely due to the fact that the fore and hindlimb development are heterochronic. We thank the reviewer for highlighting this discussion point and we now comment on this in the results section.

2. Related to this, the anterior ectopic digits are only detected in hindlimb buds, the description of the phenotype should reflect this: "The additional anterior digit in mouse hindlimb buds bears resemblance to the extra triphalangeal thumb observed in the orthologous human congenital malformations". This reviewer is of the opinion that calling it an extra triphalangeal thumb is not accurate as this phenotype is hindlimb specific in mouse.

We thank the reviewer for this comment and agree. We now add the suggested sentence to the beginning of our results section to highlight the presentation of polydactyly in mouse hindlimbs and abbreviate TT to triphalangeal toe/thumb.

3. Figure 3 (panels E-G): in comparison to the GOF SNV limb bud, development of the LOF SNV limb bud seems delayed in comparison to the GOF SNV limb bud. This needs to be clarified by accurate somite staging and/or including a comparative developmental time course as the LOF mutation is used to generate trans-heterozygous mice subsequently (e.g. Figure 4B).

The Syn 0.25 embryo in Figure 3 of our initial submission was indeed older than all other embryos in our *in situ* hybridization by a few somites. We have since repeated the *in situ* hybridization of Syn 0.25 and LOF embryos, along with controls, and updated the manuscript with somite-matched limb buds. By updating the images in Figure 3, we resolve the timing discrepancy of our embryonic limb buds. There are some morphological differences in the mice with polydactyly vs those without due to increased proliferation in the anterior limb bud caused by the ectopic *Shh*. This makes embryos with ectopic *Shh* and *Ptch1* look larger than the WT or LOF counterparts, however all animals have the same number of somites.

4. These results are unexpected or spectacular, but not “shocking”. This is the wrong term used several times in the manuscript text. For example previous studies have established that low levels of anterior-ectopic *Shh* expression are sufficient to cause preaxial polydactyly digit phenotypes in mice.

We thank the reviewer for bringing this to our attention. We did not think about this previously but can see that this is an inappropriate term and now omit this term in our manuscript.

It is amazing that SNVs cause affinity increases that result in anterior ectopic *Shh* expression. Can the authors please discuss /speculate why such a small affinity increase could result in anterior ectopic *Shh* activation rather than a posterior increase.

The expression of *Shh* is restricted to the posterior limb bud, and one contributing factor to this precise expression pattern is the presence of repressors within the anterior limb bud such as ALX4³⁴. The ancestral state of stem tetrapods is polydactylous^{35,36}, and we believe that repressors in the anterior limb bud contributed to the evolution of the five-digit state. We speculate that increased binding of activators to the ZRS could trigger transcriptional activation that overcomes the repression in the anterior limb bud, leading to ectopic expression of *Shh* in the anterior. Our updated manuscript now contains this point in the discussion section.

5. From the section entitled “Affinity-optimizing ETS SNVs across the ZRS drive GOF gene expression”, the description of the results becomes much more difficult to follow. This reviewer had to repeatedly check the 3 original manuscripts describing the datasets used to understand what was done. This criticism applies to the last 3 sections of the results. This reviewer is of the opinion, that the authors have to introduce/summarize these published studies much better before they describe their analysis. Also, all abbreviations taken over from the original studies must be clearly explained. This final part of the result section is very important but its description has to be improved such that it can stand alone, i.e. without readers having to read the other studies beforehand.

We thank the reviewer for bringing attention to this potential issue for readers. The abbreviation “MPRA” is now explained in the text. We have also expanded our introductions for the datasets from the other published studies to make this more understandable. We have also added a supplementary table to describe the enhancers used in these experiments. We hope these changes make this section readable without the need to refer to additional articles.

6. Related to this: the published datasets did not identify the French 2 and Indian 2 SNV as pathological variants with increased affinity for ETS binding, a highlight of the first part of the manuscript. To what extent does this alter the value and predictive power of the datasets used for analysis?

We think this is a great point, MPRA has limitations in its sensitivity to detect smaller effect variants, therefore variants found in MPRA datasets are typically the variants that show greater increase in expression. Additionally, spatial-temporal dynamics of expression are unlikely to be captured by analyzing a single timepoint in a cell line. As a result of these limitations, variants detected in MPRA as GOF values are likely the variants that drive the strongest expression changes. For example the ZRS French 3 variant drives strong ectopic when tested by reporter assay in the mouse limb and this variant shows significant GOF in the MPRA^{19,20}. Many variants are likely missed in MPRA studies because they lead to a subtle increase in expression, or a dynamic change that cannot be detected in these MPRA assays. Therefore, the number of affinity-optimizing SNVs that lead to GOF expression is likely higher than seen using these MPRA, and thus these MPRA assays likely underestimate the number of affinity-optimizing SNVs that drive GOF expression.

7. The robustness of GOF phenotypes is impressive in inbred C57Bl6 mice. Is there any published evidence for such robustness in different humans carrying the same SNVs? It would be great if the authors could comment on this.

We agree that the robustness of GOF phenotypes is impressive. There are a few SNVs found across different families in humans, and across different mouse lines that carry the same SNVs, and all are associated with polydactyly. We summarize this information in the table below and have described this robustness in the introduction section in the manuscript.

295	T>C	UK low-penetrant; Dutch 3	Human
404	G>C	Brazilian; French 6	Human
404	G>A	Cuban; Turkish 2; Korean	Human
406	A>G	M100081 Mouse; Thai	Mouse; human
407	T>A	DZ Mouse; 5460001	Mouse; human

Minor points:

8. Extended Data Figure 4 panel B. The legend is somewhat confusing and should be better described. This reviewer understands the sentence in question as follows: The total amount of biotinylated probe (bound and unbound fractions) is not statically different between the different samples. Please clarify.

Thank you for pointing this out. We now rephrase this sentence to read: “The total amount of bound probe relative to the unbound probe is not statistically different between French 2, Indian 2 and Syn0.25 sequences, suggesting all three sequences have the same affinity. Statistical tests performed with single factor ANOVA (p = 0.18).”

9. In the Excel file, the Supplementary Tables are not labelled.

We apologize for this oversight and thank the reviewer for catching this error. We now have labels for the supplementary tables.

Reviewer #3:

Lim and colleagues address an important problem in gene regulation, how enhancers regulate genes and how single nucleotide variants (SNVs) can change their activity. As eluded in in the intro, enhancers serve as transcription factor (TF) binding platforms that translate protein-DNA interaction into transcriptional outputs. However, how the sequence composition defines activity and how mutations will alter its function remains largely unknown. The authors decided to study this important question in one of the most well studied enhancers, the ZRS, the sole limb enhancer of Shh. Mutations in the ZRS are known to alter SHH activity in the limb thereby leading to increased and ectopic expression which in turn results in limb malformation. The activity of this enhancer has been used to study enhancer activity and gene regulation also because it is the only regulator of Shh in the limb, which appears to be rather an exception than the rule in enhancer-driven gene regulation.

The authors find that the known TF binding sites for the TF ETS have very low binding affinities as measured by Protein Binding Microarray (PBM). They also provide convincing data that this measurement reflects reality in vitro and in vivo. They have the hypothesis that mutations that increase TF binding drive pathogenic gain of enhancer function. Indeed, they can show that previously published mutations increase binding affinity by a small degree and this results in a) ectopic Shh expression and b) a limb phenotype. Furthermore, the authors produce a mutation that is predicted to increase affinity to the same degree as the other mutations and show that this change has very similar effects. In a further assay they show that an increase in binding affinity goes along with a more severe phenotype, also measured convincingly in vivo. The authors go on to test if this principle holds true in other experimental settings and other TFs. Again, they show that a gain of function in binding goes along with an increase in expression. They show that this mechanism can be used to predict the effect of SNVs on enhancer activity and pathogenicity.

Overall, this is a well written and carefully conducted study of high relevance. One of the major problems in current genetics is our lack of ability to predict the outcome of SNVs on gene expression and disease. This ms shows that very small changes can have major effects. More importantly, it shows that current models to predict biological relevant TF bindings sites may not be correct. If low affinity bindings sites are generally the more important ones, one would have to re-design the general analysis strategy of non-coding SNVs.

The strength of this paper is the in vitro analysis in combination with the careful and impressive mouse genetics together with the comprehensive follow up phenotyping analysis. Such an in vivo analysis is essential to the field which to a large extent relies on in vitro episomal assay.

Overall, the results are highly convincing and robustly demonstrate how single sequence variants can disrupt the careful balance of TF binding at enhancers to cause disease.

On the other side, the ZRS has been studied extensively and that suboptimal TF binding defines enhancer activity has been demonstrated previously. Furthermore, Shh is a rare example in which the expression domain of a gene is driven by a single enhancer. The situation may be very different in other cases where multiple enhancers drive complex expression patterns, a fact that is not discussed. Nevertheless, the authors provide additional information that allows for this type of generalization. In fact, their demonstration that other TFs such as AP1 may function in a similar manner gives important insights beyond the ZRS example.

In summary, this is a study with carefully designed in vitro and in vivo experiments that provides convincing evidence that low affinity bindings sites are important for enhancer function and that gain of function is a likely general mechanism how SNVs can cause disease.

Major comments (in no particular order)

1. There are many redundancies in the present version of the paper. Some of the intro information is repeated in the abstract, the intro and the discussion.

We thank the reviewer for this comment and have removed redundancies.

2. The discussion is too long and has many redundant parts. For example, the reader does not need another introduction about variants and enhancers.

We thank the reviewer for this comment and have removed the redundancies within the discussion which is significantly shorter.

3. The authors like the term “shockingly” – not appropriate for this type of ms.

We thank the reviewer for pointing this out, we agree it is not appropriate and have removed this term from the manuscript.

4. The authors claim that they can predict outcome based on sequence and the prediction of binding affinity (Fig. 5). Many mutations that reside within the ZRS have been described in the literature. They are associated with phenotypes of different severity (see OMIM *605522). For example, some mutations result in polydactyly only, or in a much more severe phenotype involving the lower limb similar to the mutation in Fig. 5 (e.g. VanderMeer et al. 2014). If their prediction is valid, it should also work when correlating human mutations with phenotypes.

We agree that the degree of increase in affinity should correlate with the severity of phenotype. So far we have only identified three human SNVs that increase affinity within the ZRS, these are French 2 and Indian 2 – both of which have the same affinity increase and the same phenotypes. We have also identified that the Dutch2 variant increases affinity of HOXA13 and HOXD13 binding, however as these are different TFs we are not able to compare the severity of phenotype and affinity increase across TFs. The 402 C>T (Mexican) variant from VanderMeer et al has a very severe phenotype, this is located near a cluster of Hox sites, but is not located within a site. We have not been able to determine the mechanistic cause of this variant.

5. How do the other so far reported mutations fit into the concept? The authors should show them and predict their binding affinity. Do they all result in and increase?

There are a total of 36 SNVs within the ZRS that are associated to polydactyly, of which 29 occur in humans¹⁹. None of the mutations have been attributed to an affinity-optimization in the past, as people typically look for loss or creation of TFBS. Although the ZRS is relatively well understood compared to other enhancers, only 7 of the identified human SNVs lie within validated binding sites, it is likely many of the other SNVs are within binding sites that have not yet been identified. This lack of annotation is because not all the TFs regulating the ZRS are not known and because we lack PBM datasets to identify low-affinity sites for known factors such as Hand2.

Of the seven human SNVs in known binding sites, three are in the ETS-A site, three are within Hox sites and one is in the Hand2 site. Three of these seven SNVs are affinity-optimizing SNVs. In addition to French 2 and Indian 2, we believe that Dutch 2 is caused by a Hox affinity-optimizing variant (see comment for point 4)³⁷. To investigate this, we did an EMSA and indeed see that the Dutch 2 variant leads to a greater binding of HOXA13 and HOXD13.

Two human mutations are thought to cause polydactyly due to creation of de novo ETS sites; these are the 739 A>G (US Family A&C) and 743 T>G (Australian)^{2,38}. These de novo sites fits with our model of affinity-optimization as this suggests that increased

binding of activators to the ZRS enhancer can overcome repressors to lead to aberrant activation in the anterior of the limb bud. The affinity of these de novo sites does correlate with the severity of expression via reporter assay performed in mouse limb buds as shown in Kvon et al 2020¹⁹. The AUS mutation creates a lower affinity de novo site and shows a gain in expression, while the US A&C mutation creates a higher affinity de novo site and shows a strong gain in expression¹⁹. Only a handful of individuals from the AUS and US A&C families have been studied and the majority of individuals have a triphalangeal thumb, however only in the USA A&C family is there a report of an individual with the more severe phenotype of an extra thumb^{37,39}. This suggests that the level of affinity of de novo sites may correlate with phenotype as well.

In summary, of the seven SNVs that fall within known TFBS, three can be explained by the principle of affinity-optimization, thus this mechanism explains 42% of mutations occurring within known sites. As we uncover additional binding sites within the ZRS and obtain PBM for transcription factors regulating the ZRS, we expect to discover that other human mutations are affinity-optimizing SNVs. We now include Supplementary Table 5 that annotates any known mechanisms for the 36 ZRS SNVs.

6. Duplications of the entire ZRS can also result in similar phenotypes (e.g. Wu et al. 2009). How do the authors explain this effect?

We agree it is interesting that duplication of the entire ZRS and surrounding area can lead to similar phenotypes. The smallest duplications causing polydactyly involve duplications of 16kb and larger³³. There could be several mechanisms that lead to aberrant activation of *Shh* as a result of this duplication. One possibility is that the second duplicated ZRS enhancer increases the frequency of activation of the promoter as now there are two ZRS enhancers that can contact the promoter. The duplication could act to concentrate more transcription factors within the region and thus makes it easier to nucleate transcriptional activation. Further investigation of these duplications would be beneficial to the field.

7. What is the model of the authors how an increase in TF binding can result in ectopic expression?

The expression of *Shh* is restricted to the posterior limb bud, and one contributing factor to this precise expression pattern is the presence of repressors within the anterior limb bud. For example, the repressor *ALX4* is expressed in the anterior limb bud and deletion of this repressor leads to polydactyly³⁴. We believe that the balance of activators and repressors binding to the ZRS ensures that normally the ZRS is only active in the posterior limb bud. The increased affinity of the ETS-A site tips this balance such that the activators now overcome repression in the anterior limb bud by factors such as *ALX4*. Mechanistically, we believe that the higher affinity ETS-A site allows a longer dwell time of ETS at the ETS-A site and that this may be able to nucleate formation of a functional transcriptional complex that can trigger recruitment of polymerase at the promoter. We now include present this model in the discussion.

References

1. Osterwalder, M. *et al.* Enhancer redundancy provides phenotypic robustness in mammalian development. *Nature* **554**, 239–243 (2018).
2. Lettice, L. A. *et al.* Opposing Functions of the ETS Factor Family Define Shh Spatial Expression in Limb Buds and Underlie Polydactyly. *Developmental Cell* **22**, 459–467 (2012).
3. Goodbourn, S., Zinn, K. & Maniatis, T. Human beta-interferon gene expression is regulated by an inducible enhancer element. *Cell* **41**, 509–520 (1985).
4. Klar, M. & Bode, J. Enhanceosome Formation over the Beta Interferon Promoter Underlies a Remote-Control Mechanism Mediated by YY1 and YY2. *Mol Cell Biol* **25**, 10159–10170 (2005).
5. Banerjee, A. R., Kim, Y. J. & Kim, T. H. A novel virus-inducible enhancer of the interferon- β gene with tightly linked promoter and enhancer activities. *Nucleic Acids Research* **42**, 12537–12554 (2014).
6. Thanos, D. & Maniatis, T. Virus induction of human IFN β gene expression requires the assembly of an enhanceosome. *Cell* **83**, 1091–1100 (1995).
7. Juang, Y.-T. *et al.* Primary activation of interferon A and interferon B gene transcription by interferon regulatory factor 3. *Proc Natl Acad Sci U S A* **95**, 9837–9842 (1998).
8. Tewhey, R. *et al.* Direct Identification of Hundreds of Expression-Modulating Variants using a Multiplexed Reporter Assay. *Cell* **165**, 1519–1529 (2016).
9. 100,000 Genomes Pilot on Rare-Disease Diagnosis in Health Care — Preliminary Report. *N Engl J Med* **385**, 1868–1880 (2021).
10. Lindstrand, A. *et al.* Genome sequencing is a sensitive first-line test to diagnose individuals with intellectual disability. *Genetics in Medicine* **24**, 2296–2307 (2022).
11. Farh, K. K.-H. *et al.* Genetic and epigenetic fine mapping of causal autoimmune disease variants. *Nature* **518**, 337–343 (2015).
12. Tak, Y. G. & Farnham, P. J. Making sense of GWAS: using epigenomics and genome engineering to understand the functional relevance of SNPs in non-coding regions of the human genome. *Epigenetics & Chromatin* **8**, 57 (2015).
13. Jindal, G. A. *et al.* Affinity-optimizing variants within cardiac enhancers disrupt heart development and contribute to cardiac traits. 2022.05.27.493636 Preprint at <https://doi.org/10.1101/2022.05.27.493636> (2022).
14. Zhang, W. *et al.* A global transcriptional network connecting noncoding mutations to changes in tumor gene expression. *Nat Genet* **50**, 613–620 (2018).
15. Kapoor, A. *et al.* Multiple SCN5A variant enhancers modulate its cardiac gene expression and the QT interval. *Proceedings of the National Academy of Sciences* **116**, 10636–10645 (2019).

16. Man, J. C. K. *et al.* An enhancer cluster controls gene activity and topology of the SCN5A-SCN10A locus in vivo. *Nat Commun* **10**, 4943 (2019).
17. Zhang, T., Yong, S. L., Tian, X.-L. & Wang, Q. K. Cardiac-specific overexpression of SCN5A gene leads to shorter P wave duration and PR interval in transgenic mice. *Biochemical and Biophysical Research Communications* **355**, 444–450 (2007).
18. Liu, G. X., Remme, C. A., Boukens, B. J., Belardinelli, L. & Rajamani, S. Overexpression of SCN5A in mouse heart mimics human syndrome of enhanced atrioventricular nodal conduction. *Heart Rhythm* **12**, 1036–1045 (2015).
19. Kvon, E. Z. *et al.* Comprehensive In Vivo Interrogation Reveals Phenotypic Impact of Human Enhancer Variants. *Cell* **180**, 1262-1271.e15 (2020).
20. Kircher, M. *et al.* Saturation mutagenesis of twenty disease-associated regulatory elements at single base-pair resolution. *Nat Commun* **10**, 3583 (2019).
21. Melnikov, A. *et al.* Systematic dissection and optimization of inducible enhancers in human cells using a massively parallel reporter assay. *Nat Biotechnol* **30**, 271–277 (2012).
22. Escalante, C. R., Nistal-Villán, E., Shen, L., García-Sastre, A. & Aggarwal, A. K. Structure of IRF-3 Bound to the PRDIII-I Regulatory Element of the Human Interferon- β Enhancer. *Molecular Cell* **26**, 703–716 (2007).
23. Wang, B., Fallon, J. F. & Beachy, P. A. Hedgehog-Regulated Processing of Gli3 Produces an Anterior/Posterior Repressor Gradient in the Developing Vertebrate Limb. *Cell* **100**, 423–434 (2000).
24. Hill, P., Wang, B. & Rüther, U. The molecular basis of Pallister Hall associated polydactyly. *Hum Mol Genet* **16**, 2089–2096 (2007).
25. Tiecke, E. *et al.* Expression of the short stature homeobox gene Shox is restricted by proximal and distal signals in chick limb buds and affects the length of skeletal elements. *Dev Biol* **298**, 585–596 (2006).
26. Ohuchi, H. *et al.* The mesenchymal factor, FGF10, initiates and maintains the outgrowth of the chick limb bud through interaction with FGF8, an apical ectodermal factor. *Development* **124**, 2235–2244 (1997).
27. Akiyama, H. *et al.* Misexpression of Sox9 in mouse limb bud mesenchyme induces polydactyly and rescues hypodactyly mice. *Matrix Biology* **26**, 224–233 (2007).
28. Rodriguez-Esteban, C. *et al.* The T-box genes Tbx4 and Tbx5 regulate limb outgrowth and identity. *Nature* **398**, 814–818 (1999).
29. Frayling, T. M. *et al.* A Common Variant in the FTO Gene Is Associated with Body Mass Index and Predisposes to Childhood and Adult Obesity. *Science* **316**, 889–894 (2007).
30. Dina, C. *et al.* Variation in FTO contributes to childhood obesity and severe adult obesity. *Nat Genet* **39**, 724–726 (2007).

31. Smemo, S. *et al.* Obesity-associated variants within FTO form long-range functional connections with IRX3. *Nature* **507**, 371–375 (2014).
32. de Araújo, T. M. & Velloso, L. A. Hypothalamic IRX3: A New Player in the Development of Obesity. *Trends in Endocrinology & Metabolism* **31**, 368–377 (2020).
33. Lohan, S. *et al.* Microduplications encompassing the Sonic hedgehog limb enhancer ZRS are associated with Haas-type polysyndactyly and Laurin-Sandrow syndrome. *Clinical Genetics* **86**, 318–325 (2014).
34. Qu, S. *et al.* Polydactyly and ectopic ZPA formation in Alx-4 mutant mice. *Development* **124**, 3999–4008 (1997).
35. Saxena, A., Towers, M. & Cooper, K. L. The origins, scaling and loss of tetrapod digits. *Philos Trans R Soc Lond B Biol Sci* **372**, 20150482 (2017).
36. Coates, M. I. & Clack, J. A. Polydactyly in the earliest known tetrapod limbs. *Nature* **347**, 66–69 (1990).
37. Lodder, E. *Keeping Sonic Hedgehog Under the Thumb: Genetic Regulation of Limb Development*. (2009).
38. Koyano-Nakagawa, N. *et al.* Etv2 regulates enhancer chromatin status to initiate Shh expression in the limb bud. *Nat Commun* **13**, 4221 (2022).
39. Gurnett, C. A. *et al.* Two novel point mutations in the long-range SHH enhancer in three families with triphalangeal thumb and preaxial polydactyly. *Am J Med Genet A* **143A**, 27–32 (2007).

Reviewer Reports on the First Revision:

Referees' comments:

Referee #1 (Remarks to the Author):

The authors provide a revised manuscript including a few additional experimental data.

I previously had some concerns with the approach and the broad conclusions that the authors draw from a single locus since the ZRS is in many ways unique. I had asked for validation experiments on a second independent locus.

In their rebuttal letter the authors provide some preliminary data and refer to another study in Ciona (under review at a different journal). However, the manuscript itself did not really change or improve. I feel this is really a missed opportunity, but I agree with the authors that these kinds of studies would probably take several years to complete and are indeed beyond the scope of the current manuscript.

The new eQTL data are nice and show that affinity-optimizing SNVs drive GOF expression of target genes within the endogenous locus.

Overall, I am still really excited about the study and congratulate the authors on beautiful manuscript.

Referee #2 (Remarks to the Author):

Lim et al. have carefully revised their manuscript and added significant additional analyses and data to provide additional evidence in favour of a more general causal role SNVs increasing the affinity for transcription factor underlying GOF expression

They analyse the IFN- β enhancosome using data published by others. This analysis of the IFN- β enhanceosome, i.e. enhancers with redundancy, show that SNVs increasing the affinity for a key regulator of IFN- β expression correlates well with the observed increase in expression. These and other findings support the proposal that SVNs causing an affinity increase for transcriptional regulators are linked to gain of expression variants. The authors argue that performing another detailed in vivo analysis as done for the ZRS is beyond the current study.

This reviewer agrees to the latter and is of the opinion that the additional data included strengthen the study significantly and provide evidence that the that affinity optimization of SVNs can underlie GOF in both redundant and non-redundant vertebrate enhancers.

At this stage, the authors should however down-tune their conclusions and the potential use as a general approach to identify causal variants that underlie enhanceropathies. Doing this will in no way decrease the importance and impact of this very nice study. Some suggestions:

1. Last sentence of the abstract:

"... SNVs within genomes might provide a generalizable approach..." (this is not firmly and functionally established at this stage)

2. INF- β analysis – last sentence

"... redundant enhancer, therefore indicating within reporter assays that the principle..." (the same)

3. In fact, the last sentence of the results section is phrased more cautious and well summarizes the potential general relevance of their findings

Minor points

1. There is still several obvious typos/misspellings in the text.

2. Please define the term eQTL at its first use.

3. The organization of Fig. 6 A-D is not chronological with its description in the results section. This can cause confusion. Please indicate in the Figure legend what the yellow box in the left-most panel of Fig. 6E left indicates.

Referee #3 (Remarks to the Author):

The authors have fully answered my questions. The ms has improved substantially in flow and quality.

Referee #1 (Remarks to the Author):

The authors provide a revised manuscript including a few additional experimental data.

I previously had some concerns with the approach and the broad conclusions that the authors draw from a single locus since the ZRS is in many ways unique. I had asked for validation experiments on a second independent locus.

In their rebuttal letter the authors provide some preliminary data and refer to another study in Ciona (under review at a different journal). However, the manuscript itself did not really change or improve. I feel this is really a missed opportunity, but I agree with the authors that these kinds of studies would probably take several years to complete and are indeed beyond the scope of the current manuscript.

The new eQTL data are nice and show that affinity-optimizing SNVs drive GOF expression of target genes within the endogenous locus.

Overall, I am still really excited about the study and congratulate the authors on beautiful manuscript.

We thank the reviewer for their excitement about this study and hope we and others will study this phenomena in other loci in the future.

Referee #2 (Remarks to the Author):

Lim et al. have carefully revised their manuscript and added significant additional analyses and data to provide additional evidence in favour of a more general causal role SNVs increasing the affinity for transcription factor underlying GOF expression

They analyse the IFN- β enhancosome using data published by others. This analysis of the IFN- β enhanceosome, i.e. enhancers with redundancy, show that SNVs increasing the affinity for a key regulator of IFN- β expression correlates well with the observed increase in expression. These and other findings support the proposal that SVNs causing an affinity increase for transcriptional regulators are linked to gain of expression variants. The authors argue that performing another detailed in vivo analysis as done for the ZRS is beyond the current study.

This reviewer agrees to the latter and is of the opinion that the additional data included strengthen the study significantly and provide evidence that the that affinity optimization of SVNs can underlie GOF in both redundant and non-redundant vertebrate enhancers.

At this stage, the authors should however down-tune their conclusions and the potential use as a general approach to identify causal variants that underlie enhanceropathies. Doing this will in no way decrease the importance and impact of this very nice study. Some suggestions:

1. Last sentence of the abstract:

“... SNVs within genomes might provide a generalizable approach...” (this is not firmly and functionally established at this stage)

We thank the reviewer for their suggestion, we now use “may provide a mechanistic approach ” to give the sense that this is a possibility:

Searching for affinity-optimizing SNVs within genomes may provide a mechanistic approach to identify causal variants underlying enhanceropathies.

2. INF- β analysis – last sentence

“... redundant enhancer, therefore indicating within reporter assays that the principle...” (the same)

We thank the reviewer for their suggestion, we now clarify this with the following:

Therefore, within the context of reporter assays, the principle of affinity-optimization applies to two classic enhancers and examples of a redundant and non-redundant enhancer.

3. In fact , the last sentence of the results section is phrased more cautious and well summarizes the potential general relevance of their findings

Minor points

1. There is still several obvious typos/misspellings in the text.

Thank you for spotting these, we have now corrected typos and misspellings.

2. Please define the term eQTL at its first use.

We now define the term eQTL as shown below: “To see if our findings generalize to other datasets, we analyzed an MPRA screen that tested lymphoblastoid regulatory elements and variants within these elements that were identified in an expression quantitative trait locus (eQTL) study⁴⁶.”

3. The organization of Fig. 6 A-D is not chronological with its description in the results section. This can cause confusion. Please indicate in the Figure legend what the yellow box in the left-most panel of Fig. 6E left indicates.

We now correct the figure panel labels to reflect the order in which they are mentioned in the text. We also include the following sentence in the legend of Figure 6E: Green bars indicate SNVs that cause GOF expression within analyzed MPRA datasets, yellow bar indicates SNVs that cause GOF expression in our current manuscript, namely French 2 and Indian 2.

Referee #3 (Remarks to the Author):

The authors have fully answered my questions. The ms has improved substantially in flow and quality.

We are glad we address your questions and thank you for helping us improve our manuscript.